

# Mesozooplankton structure and functioning in the western tropical South Pacific along the 20° parallel south during the OUTPACE survey (February –April 2015)

François Carlotti[1], Marc Pagano[1], Loïc Guilloux[1], Katty Donoso[1], Valentina Valdés[2,3], Olivier Grosso[1], Brian P. V. Hunt[1,4,5,6]

[1] Aix Marseille Université, Université de Toulon, CNRS, IRD, OSU PYTHEASMediterranean Institute of Oceanography, MIO, UM 110, 13288, Marseille, Cedex 09, France

[2] Millennium Institute of Oceanography (IMO), Universidad de Concepción, Concepción, Chile

[3] Programa de Doctorado en Oceanografía, Departamento de Oceanografía, Facultad de Ciencias Naturales y Oceanográficas, Universidad de Concepción, Concepción, Chile

[4] Institute of the Oceans and Fisheries, University of British Columbia, Vancouver, V6T 1 Z4, British Columbia, Canada

[5] Department of Earth, Ocean and Atmospheric Sciences, University of British Columbia, Vancouver, British Columbia,

Canada,

[6] Hakai Institute, Heriot Bay, British Columbia, Canada,

*Correspondence to*: François Carlotti (francois.carlotti@mio.osupytheas.fr)

**Abstract.**

This paper presents new results on spatial and temporal distribution patterns of mesozooplankton in the western tropical South Pacific along the 20°S south visited during austral summer (February – April 2015). By contributing to the interdisciplinary OUTPACE (Oligotrophy to UlTra-oligotrophy PACific Experiment) project (Moutin et al., 2017), the specific aims of this study dedicated to mesozooplankton observations were (1) to document zooplankton density, species diversity, and biomass along the transect, and (2) to characterize the trophic pathways from primary production to
mesozooplankton . Along a West-East transect of 4000 km from New Caledonia to the French Polynesia, 15 short-duration stations (SD-1 to SD-15, 8hours each) dedicated to a large-scale description, and three long-duration stations (LD-A to LD-C, 5days each), respectively positioned (1) in offshore northern waters of New Caledonia, (2) near Niue Island, and (3) in the subtropical Pacific gyre near the Cook Islands, were sampled with a Bongo Net with 120 µm mesh size net for quantifying mesozooplankton abundance, biomass, community taxonomy and size structure, and size fractionated content of $\delta^{15}N$.
Subsequently, the contribution of Diazotroph Derived Nitrogen (DDN (%) to zooplankton δ15N (ZDDN) values at each



station was calculated, as well as an estimation of zooplankton carbon demand and grazing impact and of zooplankton excretion rates.

The mesozooplankton community showed a general decreasing trend in abundance and biomass from West to East, with a clear drop in the ultra-oligotrophic waters of the subtropical Pacific gyre (LD-C, SD-14 and SD-15). Higher

abundance and biomass corresponded to higher primary production of more or less ephemeral blooms linked to complex mesoscale circulation in the Coral Sea and between the longitudes 170-180°W. Copepods were the most abundant group (68 to 86% of total abundance), slightly increasing in contribution from west to east while, in parallel, gelatinous plankton decreased (dominated by appendicularians) and other holoplankton. Detritus in the net tow samples represented 20 - 50% of the biomass, the lowest and the highest values being obtained in the subtropical Pacific gyre and in the Coral Sea,

respectively, linked to the local primary production and the biomass and growth rates of zooplanktonic populations.

Taxonomic compositions showed a high degree of similarity across the whole region, however, with a moderate difference in subtropical Pacific gyre. Several copepod taxa, known to have trophic links with *Trichodesmium*, presented positive relationships with *Trichodesmium* abundance, such as the Harpacticoids *Macrosetella*, *Microsetella* and *Miracia*, and the Poecilostomatoids *Corycaeus* and *Oncaea*. At the LD stations, the populations responded to the local bloom with a

large production of larval forms, reflected in increasing abundances but with limited (station LD-A) or no (station LD-A) biomass changes.

Diazotrophs contributed up to 67 and 75% to zooplankton biomass in the western and central Melanesian Archipelago regions respectively, but strongly decreased to an average of 22% in the subtropical Pacific gyre (GY) and down to 7% occurring in the most eastern station (SD-15). Using allometric relationships, specific zooplankton ingestion rates were

estimated between 0.55 and 0.64 $d^{-1}$ with the highest mean value at the bloom station (LD-B) and the lowest in GY, whereas estimated weight specific excretion rates ranged between 0.1 and 0.15 $d^{-1}$ for $NH_4$ and between 0.09 and 9.12 $d^{-1}$ for $PO_4$. Daily grazing pressure on phytoplankton stocks and daily regeneration by zooplankton were as well estimated for the different regions showing contrasted impacts between MA and GY regions. For the 3 LD stations, it was not possible to find any relationship between the abundance and biomass in the water column and swimmers found in sediment traps. Diel

vertical migration of zooplankton, which obviously occurs from observed differences in day and night samples, might strongly influence the community of swimmers in traps.



# 1 Introduction

The western tropical South Pacific (WTSP) is a vast oceanic area extending from the Coral Sea in the west to the western boundary of the South Pacific Subtropical Gyre (SPSG) in the east, and centered on the 20°S parallel. It is one of the most understudied oceanic regions in terms of the planktonic food web, despite supporting some of the largest tuna fisheries
in the world and showing variable production in response to ENSO events (Longhurst, 2006; Le Borgne et al, 2011; Smeti et al. 2015; Houssard et al. 2017).

Over the last decade the WTSP has been the subject of a number of studies concerning the biogeographical distributions of picoplankton (see Buitenhuis et al., 2012 for the data synthesis; Campbell et al. 2005) and diazotrophs (Shiozaki et al., 2014; Bonnet et al. 2015, 2017), due to their key roles in biogeochemical cycling and the functioning of
10 oligotrophic subtropical pelagic ecosystems. In this stratified oligotrophic ocean, a major source of new N for the pelagic food web appears to be $N_2$ fixation by unicellular (Zehr et al., 2001; Campbell et al., 2005; Bonnet et al. 2015) and filamentous cyanobacteria (Bonnet et al., 2009; Moisander et al., 2010; Dupouy et al., 2011). This latter form may accumulate substantial biomass after massive blooms in the summer (Campbell et al., 2005; Dupouy et al., 2011).

The contribution of blooms of cyanobacteria to the food web appears to be highly variable, and remains controversial.
(Le Borgne et al 2011). Abundances of zooplankton have been linked to blooms of *Trichodesmium* (Landry et al. 2001), but in most cases, a high biomass of cyanobacteria does not result in increased productivity of zooplankton because some cyanobacteria are toxic or unpalatable (Turner 2014). Grazing on *Trichodesmium* has been considered as a food source for only a few zooplankton species, mainly harpacticoid copepods (Hawser et al. 1992; O'Neil and Roman 1994; O'Neil 1998), however, recent studies have provided evidence of zooplankton species feeding on various types of diazotrophs. In the
Amazon River plume, copepods were shown to consume diatom-diazotroph assemblage DDAs (*Hemialus-Richelia* and *Rhizosolenia-Richelia*, diatom-diazotroph respectively), diazotrophic unicellular cyanobacteria UCYN-A Candidatus *Atelocyanobacterium thalassa*, UCYN-B *Crocosphaera watsonii*, and the colonial cyanobacterium *Trichodesmium* (Conroy et al. 2017). Recently, consumption of UCYN-C by zooplankton was observed in a mesocosm experiment performed in the oligotrophic Noumea lagoon in the southwest Pacific (Hunt et al., 2016), while the *nifH* gene, indicative of $N_2$ fixation, was
measured in the guts of zooplankton including the copepods *Pleuromamma*, *Pontella*, and *Euchaeta* in the western equatorial and subtropical Pacific waters (Azimuddin et al. 2016).

Concomitant surveys planned to identify both diazotroph blooms and zooplankton distributions are rare. The multidisciplinary ANACONDAS program (Amazon River influence on nitrogen fixation and export production in the western tropical North Atlantic) was dedicated to investigating the role of the Amazon plume in stimulating offshore
nitrogen fixation, including nitrogen supplied by nitrogen-fixing bacteria, and export production during the river's high-discharge period (May–June 2010). That study showed clear evidence of consumption of DDAs, *Trichodesmium*, and unicellular cyanobacteria by calanoid copepods (Weber et al., 2016; Conroy et al. 2017). In another recent paper, Azimuddin et al. (2016) presented data analysis to understand the diversity and abundance of potentially diazotrophic microorganisms



associated with marine zooplankton, especially copepods. That study was based on the nifH gene in zooplankton samples, mainly copepods, collected in 12 locations in the Pacific Ocean, four stations in the subarctic and subtropical North Pacific, including the ALOHA station, and eight stations in the tropical and subtropical areas of the South Pacific.

If we consult the "Copepod database" (https://www.st.nmfs.noaa.gov/copepod/) the Tropical South Pacific Ocean is among the least sampled regions in the word ocean for zooplankton investigation. The most complete ecosystem studies in the region were performed by the US (Murray et al., 1995) and French JGOFS programs (Leborgne & Landry, 2003) , in the Equatorial South Pacific (see the review by Le Borgne et al., 2002). These programs included dedicated observations on zooplankton distribution and associated fluxes (White et al., 1995; Zhang et al, 1995; Le Borgne & Rodier, 1997; Le Borgne et al, 1999; Le Borgne et al., 2003). One joint program, Zonal Flux (April l5-May 14, 1996), was an equatorial transect cruise made during a "La Nina" event (April-May 1996) in the equatorial Pacific upwelling. In the WTSP, zooplankton studies are rare and largely confined to the Coral Sea (Le Borgne et al., 2010; Smeti et al 2015). In the eastern tropical South Pacific, regular campaigns of the Scripps Institution of Oceanography in the 1960's provide information on zooplankton taxon distributions (see the review by Fernández-Álamo & Färber-Lorda, 2006).

The OUTPACE survey (Oligotrophy to UlTra-oligotrophy PACific Experiment, 18 February and 3 April 2015), aboard the RV L'Atalante, was designed specifically to sample a variety of trophic conditions along a west–east transect covering 4000 km in the SE Pacific Ocean from the western part of the Melanesian archipelago (New Caledonia) to the western were boundary of the South Pacific gyre (French Polynesia). The aims of the OUTPACE project (Moutin et al. 2017) were (1) to characterize the zonal changes in biogeochemistry and biological diversity across the western tropical South Pacific during austral summer conditions; (2) to quantify primary production and fate of organic matter (including carbon export) in three contrasting trophic regimes with increasing oligotrophy, with a particular emphasis on the role of dinitrogen fixation in areas of *Trichodesmium* blooms; and (3) to obtain a representation of the main biogeochemical fluxes and dynamics of the planktonic trophic network.

The primary aims of the present study dedicated to mesozooplankton observations were (1) to document zooplankton density, species diversity, and biomass along the transect, and (2) to characterize the trophic pathways from primary production to mesozooplankton, and in this way to contribute to these three main objectives.

## 2 Material and methods

### 2.1 Study site and sampling strategy

The OUTPACE survey was performed aboard the RV L'Atalante during austral summer conditions between 18 February and 3 April 2015 in the WTSP Ocean from New Caledonia (western part of the Melanesian Archipelago) to the French Polynesia, along a West-East transect covering *ca*. 4000 km between the latitudes 17°S (station SD-11) and 22°S (station SD-5) (**Fig. 1**). This region is impacted by the El Niño–Southern Oscillation (ENSO), known to be the most




important mode of SST variability on interannual to decadal timescales (Sarmiento and Gruber, 2006). The year 2015 was classified as an El Niño event, which was reflected on SST and chl-*a* satellite data (Moutin et al., 2017). Along this transect, two types of stations were sampled (**Fig. 1**): 15 short-duration stations (SD1 to SD15, 8 h) dedicated to a large-scale description, and three long-duration stations for Lagrangian process studies, respectively LD A station (19°12.8'S - 164°41.3'E, 25 Feb.-2 Mar.) positioned in western Melanesian archipelago  waters in the western part of the transect offshore New Caledonia, LD B station (18°14.4'S - 170°51.5'W, 15-20 Mar.) in the eastern part of  Melanesian archipelago waters near Niue Island, and LD C station (18°25.2'S - 165°56.4'W, 23-28 Mar.) in the eastern part of the transect, in the subtropical Pacific gyre near the Cook Islands. All general characteristics of the stations are presented in Moutin et al. (2017, their Table 1).

Real-time-satellite images (altimetry, SST, ocean color) combined with  drifter trajectories initiated during the first part of the cruise were used to define the best positions of these three stations using two criteria: sea surface chlorophyll levels to characterize the main sampled regions and minimal current intensity in each region to increase the chance of sampling an homogeneous water mass (Moutin et al, 2017; De Verneil et al., in review). LD-A and LD-B stations were characterized by local maxima of sea surface chl-*a* to sample the Melanesian Archipelago zone whereas chl*a* minima characterized LD-C representing typical waters of the subtropical gyre.

## 2.2 Mesozooplankton sampling

Zooplankton collection was conducted at 14 of the SD stations (the station SD-13 was not sampled for zooplankton) and at the 3 LD stations, which were sampled once during the day and once during the night for each of the 5 days of station occupation. Sampling was done with a Bongo Net (70 cm mouth diameter) with 120 μm mesh nets mounted with filtering cod-ends. The nets were equipped with Hydrobios flowmeters. Hauls were done from 200m depth to the surface at a speed of 1 m.s$^{-1}$. One of the cod-ends was used for biomass measurements. The second one was preserved in 4% buffered formaldehyde for later taxonomic identification, abundance and size spectrum analyses. Volume filtered by the nets (V) was calculated using the formula: V= R * S * K, combining  the flowmeter counts (R), the mouth area of the net (S=0.38m$^2$), and the pitch of the impeller of the flowmeter (K) provided by the manufacturer, and equals to 0.3 m/revolution.

## 2.3 Dry weight measurement

The biomass sample was processed onboard. Just after collection, each sample was filtered onto a pre-weighed GF/F filter (47 mm) and oven dried at 60 °C for 2 days.  The average biomass concentration (in mg DW m-3) in the upper 200 m was calculated from the zooplankton dry weight (mg), obtained as difference between the weight of the filter + sample and the weight of the filter, and the water column sampled volume.




### 2.4. Identification, abundance and individual size and weight of the zooplankton taxa

The taxonomic composition was determined for each formalin sample. Samples were split using a Motoda box, and at least 100 individuals of the more abundant taxa were counted in each sub-sample under a dissecting microscope, a LEICA MZ6. Species/genus identification was made according to Rose (1933), Tregouboff and Rose (1957) and Razouls et al (2005-2017). The abundance of the various taxa (groups, genera or species) was divided by the sample volume to get concentration of individuals per cubic meter (ind.m$^{-3}$). The diversity of the zooplankton was determined using the Shannon-Weaver index (Shannon and Weaver, 1949).

Approximates of the individual size (total length) and relative dimensions (length/width) of the different taxa were computed from literature values: summarized data for copepod species in Razouls et al. (2005-2017), mean size values of the other taxa from Tregouboff and Rose (1957) and Conway et al (2003).

For comparison with Zooscan results (see below), we computed the area of each taxa (A) from its dimensions to calculate its equivalent circular diameter (ECD):

$$ECD = \sqrt{\left(4 * \frac{A}{\pi}\right)}$$

We also estimated individual dry weight (DW) using the area-dry weight relationships obtained by Lehette and Hernandez-Leon (2009) for subtropical copepods and mesozooplankton.

### 2.5 Abundance, biomass and size structure using the Zooscan

Samples were digitized with the ZooScan digital imaging system (Gorsky et al., 2010) to determine the size structure of the zooplankton communities, as detailed in Donoso et al (2017). Each sample was divided into 2 fractions (<1000 and > 1000μm) and each fraction was then split using a Motoda box until it contained approximately 1000 objects. The resulting samples were poured onto the scanning cell and zooplankton organisms were manually separated with a wooden spine in order to avoid overlapping organisms. After scanning, each image was processed using ZooProcess using the image analysis software Image-J (Grosjean et al., 2004; Gorsky et al., 2010). Only objects having an equivalent circular diameter (ECD) of > 300μm were detected and processed. Finally, Plankton Identifier (http://www.obs-vlfr.fr/~gaspari/Plankton_Identifier/index.php) was used for automatic classification of zooplankton into 12 categories. Among them two categories of non-zooplankton organisms, aggregates and fibers were grouped as detritus. A training set of about 1000 objects selected automatically from different scans was used to discriminate and classify between organisms, aggregates and fibers. Afterwards, each scan was corrected using the automatic analysis of images.

Zooplankton abundance estimated from Zooscan (ind m$^{-3}$) was calculated from the number of validated vignettes in Zooscan samples, taking into account the scanned fraction and the sampled volume from the net tows. Zooplankton estimated dry weight of each vignette was calculated from its area using the regression equation obtained for mesozooplankton by Lehette and Hernández-León (2009).



Below, the terms "ZOOSCAN abundance" and "ZOOSCAN biomass" will indicate values derived from the laboratory ZOOSCAN processing. The abundance and biomass of organisms were first calculated for four size fractions (< 500, 500–1000, 1000–2000 and > 2000 μm) based on their ECD, and then summed to deliver the total average abundance and biomass per sample over the upper 200 m.

## 2.6 Stable isotope analysis

Nitrogen isotope ratios ($\delta^{15}$N) were measured for the zooplankton size fractions collected for biomass measurement and for particulate organic matter (POM) samples collected at 5 m depth at each station. Zooplankton samples were first homogenized using a mortar and pestle, and packaged into ~ 1 mg sub-samples. Stable isotope analysis was performed with

10 an Integra CN, SerCon Ltd. EA-IRMS. $\delta^{15}$N values were determined in parts per thousand (‰) relative to the external standard of atmospheric N.

The mean $\delta^{15}$N value for each station was calculated as the mean of all size fractions, weighted by size fraction biomass. Subsequently, the contribution of Diazotroph Derived Nitrogen (DDN (%) to zooplankton $\delta^{15}$N (ZDDN) values at each station was calculated using a two source mixing model following (Sommer et al., 2006):

$$\% \, ZDDN = 100 * \left( \frac{\delta 15 N_{zpl} - \delta 15 N_{zplref}}{TEF + \delta 15 N_{diazo} - \delta 15 N_{zplref}} \right)$$

where $\delta^{15}$N$_{zpl}$ is the isotopic signature of the zooplankton collected;. the trophic enrichment factor (TEF) was set at 2.2 ± 0.3 ‰ (McCutchan et al., 2003; Vanderklift and Ponsard, 2003); the isotopic signature of diazotrophs ($\delta^{15}$N$_{diazo}$,) was set in a range of -1 to -2 ‰ (Montoya et al., 2002); the isotopic signature of zooplankton assuming nitrate based

20 phytoplankton production ($\delta^{15}$N$_{zplref}$), was set at 6 ‰ for the Melanesian Archipelago stations - a value calculated for the ocean west of New Caledonia where nitrogen fixation is reduced (Hunt et al., 2015) – and at 10.73 ‰ for the SPG samples - the mean value of POM samples in the SPG + 2.2 ‰ trophic enrichment for the primary consumer level-. Minimum, average and maximum % ZDDN were estimated using the lower, mean and upper bounds of TEF and the $\delta^{15}$N$_{diazo}$ values cited above.

## 2.7 Other data from OUTPACE survey used for interpretation and comparison

Acquisition of environmental data used in the present paper is presented in different companion papers. Briefly, temperature, salinity, and density were collected with a CTD SeaBird SBE 9 and particle distribution with a UVP, both mounted on a rosette (de Verneil et al, this issue), whereas chlorophyll-a and pheophytin concentrations were estimated for

30 different depths from bottle water samples using the fluorometric method (Dupouy et al, this issue). The depth of the mixed



layer (MLD) was calculated using a threshold density deviation of 0.03 kg m$^{-3}$ from the value at a reference depth (de Verneil et al, this issue).

*Trichodesmium* colonies abundance was estimated using three methods: a SatlanticMicroPro free-fall profiler equipped with the OCR-504S UV-VIS radiometer, a UVP5, and from backscattering and absorption coefficients Dupouy et

al. (this issue). Chl-*a* contained in *Trichodesmium*, nanoeucaryotes, and micro and macroalgae were estimated from HPLC measurements. Abundance and distribution of cyanobacterial diazotrophs were estimated by Stenegren et al (this issue) from microscopic analyses of >10 μm cyanobacterial diazotrophs and results of 'at sea' qPCR analyses of four unicellular diazotrophic targets (UCYN-A1, UCYN-A2, UCYN-B and UCYN-C). Primary productivity was determined using a $^{14}$C labelling method according to Van Wambeke et al (this issue)

A drifting array equipped with three PPS5 sediment traps and various captors was deployed at each LD station for 5 days at 3 depths (see Caffin et al., 2017 for details). Swimmers found in the trap were quantified and genera identified, and weighted. Zooplankton C (Zoo-C), N (Zoo-N) and P (Zoo-P) mass measured at each depth of each station. Only data from the sediment trap situated at 150 m deep were used here.

**2.8. Estimation of zooplankton carbon demand and grazing impact and of zooplanktonexcretion rates**

The zooplankton carbon demand (ZCD in mgC m$^{-3}$ d$^{-1}$) was computed based on estimates of biomass and of ration:

$$ZCD = Ration\ B_{zoo}$$

where $B_{zoo}$is the biomass of zooplankton in mgC m$^{-3}$, and *Ration* (d$^{-1}$) is the amount of food consumed per unit of biomass per day, calculated as:

$$Ration = (g_z + r) / A$$

where $g_z$ is the growth rate of zooplankton, $r$ is the weight specific respiration and $A$ is assimilation efficiency.

$g_z$ was calculated following Zhou et al. (2010):

$$gz(w, T, C_a) = 0.033 \left( \frac{C_a}{C_a + 205 e^{-0.125T}} \right) e^{0.09T} w^{-0.06}$$

as a function of sea water temperature ($T$, °C), food availability ($C_a$, mgC m$^{-3}$, estimated from Chl-*a*), and weight of zooplankton individuals ($w$,mgC).

Following Nival et al. (1975), we considered a constant values of $A = 0.7$ d$^{-1}$. For respiration we applied a constant values ($r$=0.27 d$^{-1}$) derived from the mean results of McKinnon et al (2015) for Australian coastal zooplankton.

ZCD were thus estimated for each taxa and then summed to estimate ZCD of total zooplankton. To estimate the potential clearance of phytoplankton by zooplankton, we compared ZCD to the phytoplankton stock, converted to carbon assuming a classical C:Chl-*a* ratio of 50:1, and to the phytoplankton primary productions estimated by Van Wambeke et al

(this issue).

Ammonium and phosphorus excretion rates were estimated for each taxa and station from the multivariate regression equations by Ikeda (1985) in which independent variables are animal body weight (carbon) and temperature. The daily $NH_4$ and $PO_4$ excretion values by total zooplankton equal the sum of values for all taxa. To estimate the potential contribution of zooplankton excretion to N and P requirements for phytoplankton, we estimated the latter from primary production using

Redfield's ratios.

### 2.9 Data analysis

Principal component analysis (PCA) was used to explore spatial patterns of the environmental variables temperature, salinity, chlorophyll-a, percentage of chlorophyll a (ratio Chl-*a* / Chl-*a* +Phae) using average values between 0-200m depth,

to be consistent with the net haul depth. The depth of the mixed layer (MLD) was included in the calculations. The data were normalized before the analyses run using the Primer 6.0 software.

Two-way analyses of variance were run on the dataset concerning the LD station, to explore the differences between day and night samples and between stations for the zooplankton parameters.

Spearman's rank-correlations (Rs) were computed to test relationships between zooplankton variables and

environmental parameters.

To better understand the changes in the community structure, Rank Frequency Diagrams were constructed by plotting the ranks of all species on the x-axis (in decreasing order of frequency) against their frequency value on the y axis. Diversity was calculated for zooplankton and copepod taxa using the Shannon–Wiener diversity index.

Spatial variations of the zooplankton community composition were investigated using multivariate analysis,

specifically Nonmetric Multidimensional Scaling (NMDS). A matrix species - stations of square root transformed abundance data. was used to estimate station similarity using the Bray Curtis metric. The similarity matrix was then ordinated using NMDS. A SIMPER (percentage of similarity) analysis was performed to identify the species contributing most to similarity or dissimilarity between stations for the station groups identified by NMDS.

Finally, to investigate which environmental variables were most strongly related to community composition we used

a multiple linear regression in which the first two dimensions of the NMDS analysis are the independent variables while the environmental variables are considered the dependent variables (Hosie and Cochran, 1994).

The used environmental variables are the same as used in the PCA and we also considered the abundance of *Trichodesmium*, derived from Stenegren et al. (this issue). Analyses were run using Primer 6 for PCA and NDMS and with Statistica v.6 for ANOVA, regression and correlation



## 3 Results

### 3.1 Hydrology and trophic conditions along the transect

In the PCA of environmental data, the first two axes explained 70% of the total variance of which 50% was accounted for by the first axis (**Fig 2**). The first axis clearly separated the Subtropical Gyre (GY) stations (stations LD-C, SD-14 and SD-15), characterized by low chlorophyll but high temperature, salinity and MLD values, from the stations of the Melanesian Archipelago (MA). The second axis opposed two clouds of stations within this latter group: the first included the western stations close to Noumea and the Loyalties (W-MA) and LD-B sampled in "blooming" condition (called BL) and characterized by higher percentage of chlorophyll *a* to total pigments (>67%); and the second cloud (57±0.09% Chl-*a*) grouped the stations referred to as central and eastern MA stations (CE-MA). Mean values of environmental data in each cloud are given in **Table 1**. Salinity was significantly lower in NA than in GY and BL (ANOVA; $p<0.05$), temperature was significantly lower in MA than in GY, and MLD was significantly deeper in GY than in W-MA and CE-MA ($p<0.05$). Chl-*a* was significantly lower in GY than in the three other zones and % Chl-*a* was significantly higher in W-MA and BL than in GY and CE-MA.

### 3.2 Spatial variations of zooplankton abundances

The total zooplankton abundance estimated with microscope counting varied from 409 to 2017 ind m$^{-3}$ (**Fig. 3A and Table 2**). Highest values, but high variability as well, was observed in the New Caledonia region (SD1 to 4, and LD-A) and a general decreasing trend from West (station SD-4) to East (station SD-15), with local increases linked with Chl-*a* increase. There was a clear drop in abundance in GY stations (LD-C, SD-14 and SD-15) compared to all the other zones (W-MA, CE-MA and BL; $p<0.05$). The contribution of the size-fraction > 300μm ECD varied from 253 to 1140 ind m$^{-3}$ representing 49 to 63% of the total microscope counted zooplankton abundance (**Fig 3A and Table 2**). Microscope abundance showed relatively good agreement with ZOOSCAN abundance (Rs=0.627, p=0.007). The ratio of abundance of zooplankton size fractions above and below 300μm ECD did not show any spatial trend. Zooplankton abundance was negatively correlated with water column temperature (Rs=-0.511, p= 0.028) and MLD (Rs=-0.790, p=0.000). It was positively correlated with Chl-*a* (Rs=0.498, p=0.042) when considering all of the transect stations, but the correlation was negative for stations in the New Caledonia region (SD1 to 4, and LD-A; Rs=-0.900; p=0.037) and highly positive for other stations (SD-5 to SD-15, LD-B and LD-C; Rs=0.804; p=0.002).

Copepods was the most abundant group (68 to 86% of total abundance) with a slight increase of their contribution from west to east (**Fig. 3B**) with in parallel a slight decrease of other contributors (gelatinous plankton and other holoplankton). Thus, copepod dominance was more prominent in the GY zone (79 to 86%) than in the other sites (<80%). Gelatinous zooplankton represented 7.5 to 21% of zooplankton abundance with lowest contributions in stations LD-C and SD-15 in the GY zone. This group was highly dominated by appendicularians (8-17%) whereas siphonophores, doliolids,



salps and hydrozoans represented <0.5% of the total zooplankton abundance. Chaetognaths represented up to 3.8% (mean = 2.1%) of abundance and were particularly rare in the GY zone (<1%) and at SD-1 (0.2%). Other holoplanktonic taxa (2.3-12.7%) included mostly Thecosomata (1.2-10.2%) Ostracods (1-4%) and Euphausiids (<1%). Meroplankton included mostly polychaete larvae (0.2-0.5%) and lamellibranch larvae (0.1-0.4%), which were present in the 4 zones.

5        Among copepods, larval forms were dominant (69-88% of copepod abundance) and included mostly copepodites (42-82%). In the GY zone the proportion of adults (mean=15 ± 2%) was lower than in the 3 other zones (mean>18%), whereas the percentage of adult females was the lowest in W-MA.

### 3.3 Spatial variations of zooplankton, biomass and size structure

10        As for abundance, the total zooplankton biomass estimated by cumulating estimations from microscope counting (fraction <300 µm) and Zooscan (fraction >300 µm) showed a general decreasing trend from west (SD-1) to East (SD-15), with local increases sometimes linked with Chl-*a* increase (**Fig. 4A and Table 3**). Detritus (estimated with Zooscan) were also particularly abundant (40-50%) in the Coral Sea region.

        Excepted for stations SD-2, SD-3, and SD-9, total dry weights estimated from the biovolumes of counted organisms and particles (from binocular for ECD <300 µm, and from ZOOSCAN for ECD > 300 µm) showed a good correspondence to measured total dry weight (Rs=0.721, p=0.001). The biomass fractions of ECD < 500 µm (**Fig. 4B and Table 3**) were relatively stable along the west-east transect in W-MA, CE-MA and BL both in absolute values (between 1.1 and 2 mg DW m$^{-3}$) and in percentage of total zooplankton biomass (around 20%), but slightly dropped in biomass values in GY (between 0.6 and 1.0 mg DW m$^{-3}$) with however a higher percentage contribution to the total zooplankton biomass (42% of total zooplankton biomass).

        Dry weight (weighed) as well as Zooscan zooplankton biomass were both positively correlated with Chl-*a* (RS=0.588, p=0.013 and Rs=0.68, p=0.002 respectively). As for abundance better correlations were found when considering stations SD-5 to SD-15 and LD-C, (RS=0.783, p=0.002 for both variables), whereas negative correlation was found with Zooscan zooplankton biomass for stations of the Coral Sea (Rs=-0.900, p=0.035).

25        Interestingly, detritus biomass was also well correlated with Chl-*a* when considering the whole transect data (Rs=0.721, p=0.001) and data from stations outside the Coral Sea (Rs=0.755, p=0.004).

### 3.4 Taxonomic diversity in the different oceanic regions along the transect

        From the 120 µm mesh size Bongo net, 66 zooplankton taxa were identified (**See Supplementary Table 1**) with 41 genera/species of copepods plus miscellaneous nauplii and copepodites). Copepods constituted the bulk of the zooplankton community abundance with 76.7% (SD=4.4 %) of the counted organisms over the whole area, and copepodites represented a little more than half of the counted copepods (mean=65.3 %, SD=10.0 %). Among copepods *Clausocalanus/Paracalanus*



(25% of copepod abundance), *Oithona* (19%), *Oncaea* (18%), *Corycaeus* (7.6%) and *Microsetella* (4.6%) were the most abundant genera and were present in all stations sampled. All these copepod taxa were completed in the top ten species in frequency abundance for the 4 regions (**Table 4**) with appendicularians, thecosomata, chaetognaths (except GY).

The mean rank frequency distributions of the total zooplankton community for the four regions showed very similar patterns at W-MA, CE-MA and BL zones (**Figure 5**) with the dominance of the 5 same taxa (*Clausocalanus*/*Paracalanus*, *Oithona*, *Oncaea*, appendicularia and nauplii, representing each > 9% abundance) and then a progressive decrease in abundance of the following rank species (4-6% at rank 6 to ca 1% at rank 14-15) giving regular convex shapes of the diagrams. The mean distribution for the GY zone was distinctly different with the appearance of *Corycaeus* at rank 3 (11% abundance) and the shift of nauplii from rank 5 to rank 6 and then a sharp decrease in abundance between rank 6 (8.5%) and rank 7 (2.2%). The total number of zooplankton taxa per sample varied from 25 to 40 and the Shannon index between 3.3 and 3.76 bit ind$^{-1}$ (**Table 5**). These two variables displayed their minimal mean values in the GY zone.

The mean taxonomic composition per region within the different size classes is presented in **Figure 6**. The smallest size class (<300 μm); was characterized by the very high relative abundances of *Clausocalanus* and *Paracalanus* copepodites in the GY zone (58%) compared to the other zones (35-40%). In this smallest class, nauplii (15-25%), *Oncaea* copepodites (15-30%) and, to a lesser extent, Thecosomata and *Microsetella* (5-10%) were also rather well represented in the different regions. Small copepod adults (*Oncaea*, *Oithona*, *Corycaeus*, *Clausocalanus*/*Paracalanus*) constituted the bulk of the 300-500 μm size-class, also characterized by the significant contribution of *Macrosetella gracilis* in the BL zone, and the higher relative abundance of *Corycaeus*/*Farranula* in the GY zone (32%) compared to the other zones (10-20%). The 500-1000μm class had a very similar composition between the 4 regions and was dominated by appendicularians (55-65%). The two largest size-classes (1000-2000 μm and >2000 μm) were both characterized by a strong variability between zones. The 1000-2000μm size class was highly dominated by Chaetognaths (>60%) in the W-MA and CE-MA zones, by *Pleuromamma* (56%) in the GY zone, and by Chaetognaths (48%) and *Subeucalanus* and *Calanus* copepodites (20%) in BL. The size class >2000μm presented the strongest heterogeneity in taxonomic composition between zones with dominance of salps in GY, of doliolids in BL and of siphonophores in W-MA and CE-MA. Teleost eggs and juveniles of euphausiaceae were also found in CE-MA.

Several copepod taxa, also reported in our study, have been shown to have trophic links with *Trichodesmium*: the harpacticoids *Macrosetella*, *Microsetella* and *Miracia* (see O'Neil& Roman, 1994), the Poecilostomatoids *Corycaeus* and *Oncaea* (Dupuy et al. 2016). In the present study, we found a clear link between the abundance of harpacticoids and the abundance of *Trichodesmium* with a clear decrease of both in the GY stations (**Fig 7A and Table S1**). However, only *M. gracilis* showed a significant relationship with *Trichodesmium* (Rs=0.684, p=0.002). Furthermore, *Macrosetella gracilis* and *Miracia efferata* had their highest abundance in the BL station, whereas *M. efferata* was totally absent from the GY stations. Among the Poecilostomatoids (**Fig 7B**), *Oncaea* showed a positive relationship with *Trichodesmium* (Rs=0.484, p=0.049) and displayed its highest absolute and total abundances in BL (**Table S1**), whereas *Corycaeus* had its highest relative abundance in GY and showed no significant correlation with *Trichodesmium* (p=0.900). As Azimuddin et al (2016) indicated



that *Pleuromamma*, *Pontella*, and *Euchaeta* were the major copepod genera hosting dinitrogen fixers, their spatial distributions are presented as well (**Fig. 7C**), although no significant correlation with *Trichodesmium* were found. Among the other taxa only Thecosomata, showed a positive correlation with *Trichodesmium* (Rs=0.631, p= 0.007) and displayed significant lower abundance in GY compared to W-MA, CE-MA and BL.

The NDMS ordinations based on the relative abundance of the zooplankton taxa discriminates GY stations from the other stations (**Fig 8A**, dissimilarity = 20%) mainly due to the contributions of *Corycaeus* (7%) and *Clausocalanus/Paracalanus* (6.3%) positively correlated to GY and of Appendicularians (5.6%) and Chaetognaths (5%) correlated to the other stations (**Fig. 8B**). However, the analysis did not discriminate groups among W-MA, BL and CE-MA stations, despite these groups being distinguishable on the basis of environmental data. Multiple regression analysis showed that the environmental variables that were most strongly linked to the species abundance distribution (represented by the first two dimensions of the NMDS) were *Trichodesmium* abundance (r=0=814, p=0.000), MLD (r=0.716, p=0.006), and Chl-*a* (r=0.590, p=0.05), whereas temperature and salinity had no significant contribution (p>0.3).

### 3.5. Temporal dynamics of zooplankton at the three long duration stations and comparison with sediment trap content.

In LD-A there was an increase in zooplankton abundance and a clear change in zooplankton taxonomic composition between day 1 and day 5 with an increase of the percentage of copepods but a relative decrease of gelatinous plankton (**Fig. 9A**). In the two other long stations (**Fig. 9B and 9C**), LD-B and LD-C, the abundance and taxonomic composition remained rather homogeneous with time. For the three stations, no trends were observed in the biomass values, with total biomass and size fraction specific biomasses remaining relatively consistent over time.

In LD-A, the increase of abundance without strong increase in biomass was linked with an increase of larval forms (nauplii and copepodites mainly) benefiting from the increase in Chl-*a* (Fig. 9D). At the same station, there was a high biomass of detritus with increasing percentage over time might be linked to the zooplankton population growth. The stability in LD-C of both abundance and biomass was parallel to the stability of Chl-*a*, whereas in in LD-B zooplankton did not respond to the crash of the bloom within the last two days (**Fig. 9E and 9F**).

The zooplankton biomass clearly increased in night samples compared to day samples in LD-A and LD-B whereas there was no clear diel pattern in LD-C. The organisms of the size fraction > 2000 µm (euphausids, large copepods,etc.) were the main contributors of this diel variation (**Fig. 9D to 9 F**). In LD-C, the diel variation of abundance in the upper 200 m could be observed at the taxa level for smaller forms, such as appendicularians, which did not impact the biomass.

In the sediment traps situated at 150 m, there was a more important relative contribution of copepods in LD-C compared to LD-A and LD-B, as observed in the water column (**Fig 10**). Conversely appendicularians were a large contributor of the swimmers found in the LD-C trap compared to their frequency in the water column, and by comparison to their respective frequencies in LD-A and LD-B. In LD-B, there was a sharp decrease of swimmers over time in the traps

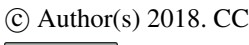

mainly due to copepods, but a relative increase of Ostracods. Pteropods had high relative contribution in the traps (20-30%
in LD-A, around 10% in LD-B and LD-C), whereas their relative abundance in the water column was low (1-4%).

### 3.6 Estimation of the trophic interactions between mesozooplankton and phytoplankton

Biomass weighted zooplankton $\delta^{15}N$ were lower in the regions W-MA, CE-MA and BL averaging 2.7‰ in W-MA
and 2‰ in CE-MA (**Figure 11A**) than in the GY were zooplankton $\delta^{15}N$ values averaged 8.5‰. The $\delta^{15}N$ values of the
zooplankton corresponded with those of the POM, these being lower west of the GY and increasing in the GY. We estimated
that diazotroph derived nitrogen contributed an average of 67 and 75% to zooplankton biomass in the W-MA and CE-MA
regions respectively (**Figure 11B**). In the GY, the diazotroph contribution to zooplankton biomass decreased to an average
of 22%, and showed a declining trend from west to east with the lowest value of 7% occurring at SD-15.

Weight specific ingestion, NH4 and PO4 excretion rates determined from allometric relationships (see Material and
Methods) and their impacts on the phytoplankton (ZCD, contribution of zooplankton regenerated nutrient to PP) are
presented in **table 6** for the different regions and long duration stations. The zooplankton ingestion rates varied between 0.55
and 0.64 d$^{-1}$ and had highest mean value in BL station and lowest in GY. The daily grazing pressure by zooplankton
represented between 3.6 and 21% of the phytoplankton stock and between 16 and 234% of the primary production, with
mean impacts varying between zones. Weight specific excretion rates varied between 0.1 and 0.15 d$^{-1}$ for NH$_4$ and between
0.09 and 9.12 d$^{-1}$ for PO4. Daily regeneration by zooplankton represented between 15 and 165% of phytoplankton needs for
N and between 3 and 34% for P. Mean lowest impacts of zooplankton on phytoplankton, both in terms of grazing and
regeneration was found in W-MA compared to other zones.

## 4 Discussion

### 4.1 What the OUTPACE transect contributes to the characterization of the two provinces ARCH and SPSG defined by Longhurst (2006)

The OUTPACE campaign delivered a unique zonal transect in the South western Tropical Pacific straddling the 20°S
latitude. This transect spanned two regions previously defined by Longhurst (2006): the south eastern part of the Archipelago
Deep Basins Province (ARCH), a province of diverse basins of the Indo Pacific archipelago of which the Coral Sea visited
during OUTPACE is the largest one; and the north western part of the South Pacific Subtropical Gyre Province (SPSG).
Along the 20°S parallel, the transition between the two regions during OUTPACE was estimated to be west of Niué Island,
between the LD-B and LD-C stations (Moutin et al. 2017). The LD-C station was performed in a cyclonic eddy in the most
oligotrophic part of the OUTPACE transect close to the Cook Islands and our PCA grouped it in a cloud of stations with SD-
14 and SD-15 (GY) which clearly belong to the SPSG region. The position of LD-B relative to the region ARCH or SPSG is





more debatable. LD-B was situated east of the Tonga trench, whereas SD-12 was just north of the trench and SD-11 west of Tonga Island with a bottom depth of 2500m. The PCA situated the LD-B station between LD-C (GY group) and SD-12 (CE-MA group) on the first axis (see Fig. 2) but the high Chl-$a$ values in LD-B excluded this station from GY but from CE-MA as well. The position of LD-B was chosen on board, the survey strategy being modified by the development of tropical cyclone Pam, and was further east than initially planned. Thus it is possible that at this latitude (20° S), the position of the limit between ARCH and SPSG is west of LD-B, at the level of Tonga Trench, and that LD-B presented special conditions due to the storm in the most western part of SPSG.

As mentioned by Longhurst (2006), the ARCH province is a mosaic of different regions. During OUTPACE, two sub-regions were differentiated by PCA (Fig. 2). The first, W-MA, in the North of New Caledonia (SD-1 to 3 and LD-A – MAW in Moutin et al., submitted) and the second, CE-MA, through the tropical island nations east of New Caledonia (eastern part of the Coral Sea), south of Vanuatu and Fiji, and north as far as Tonga (SD-4 to SD-12 - MAE in Moutin et al., submitted). The limit between the two regions in the Coral sea is linked to circulation (seasonal variation of the jet) and bathymetry. During OUTPACE, the two stations SD-4 and SD-5, at the frontier zone north of the Norfolk ridge and upon the New Hebrides trench (Ceccarelli et al 2013) and under the influence of the South Fiji jet, were grouped with the CE-MA station in our PCA (Fig. 2).

## 4. 2. Spatial structure of zooplankton biomass and abundance during OUTPACE related to physical and biogeochemical environment

The distribution of mesozooplankton abundances and biomasses during OUTPACE presented a decreasing West-East gradient over the 4000 km transect along the 20°S parallel. The pattern followed the Sea surface chlorophyll gradient, which in turn reflected the oligotrophic gradient with higher values obtained at W-MA, lower values at GY and intermediate values at CE-MA (Moutin et al., submitted). We found a good correlation between the zooplankton biomass (or abundance) and Chl-$a$ from stationsSD-5 to SD-11, but an inverse correlation in the Coral Sea stations SD-1 to SD-4).

During OUTPACE, the highest zooplankton biomass, was found in the CE-MA region (LD-A, SD4 and SD5), but high values were also found at LD-B, and to a lesser extent at SD-9. In all cases these higher values were associated with productivity enrichment linked to mesoscale features (Rousselet et al. this issue, their Fig; 3, top panel). The survey path between stations SD1 to SD5 passed through a succession of cyclonic and anticyclonic eddies, but the distance between sampling stations was unfortunately not sufficient to map them. A few studies have related the impact of mesoscale structure on zooplankton distribution in the region (Le Borgne et al. 1985; Smeti et al, 2015). Around Mare (the southernmost Loyauté Island), Le Borgne et al. (1985) found similar zooplankton enrichment not correlated with chlorophyll increase, associated with diverse mesoscale processes and in particular the island mass effect leeward (west) of Mare. It can be expected that such patterns are general features in regions where zooplankton aggregations occur more in flow-disturbed region than in free streams of jets (Rissik et al., 1997). The cyclonic or anticyclonic nature of the eddy influences not only the distributions of zooplankton abundance and biomass from the center to the edge of the eddy, but the taxonomic distribution as well





(Riandey et al. 2005). Generally, frontal aggregation of zooplankton biomass can occur at anti-cyclonic eddies peripheries (Riandey et al., 2005; Smeti et al., 2015). In oligotrophic regions, eddies potentially represent highly productive areas for zooplankton, positively impacting the biomass of all subsequent levels of the food web (Rissik & Suthers, 2000; Smeti et al., 2015). Other mesoscale activities were observed in the CE-MA region during OUTPACE between 170-180°W (Rousselet et

al. this issue) which could explain the relative increase in zooplankton biomass at SD-9.

The LD-B station was selected because of a large surface chl-*a* signal observed by satellite (de Verneil et al., 2017b) for several weeks prior to sampling, and its sampling occurred at an advanced bloom stage with high N2 fixation rates as the source of new production (Caffin et al., 2017). As a consequence of the late stage at which this bloom was sampled, the potential physical processes that induced its formation cannot be definitively established (de Verneil et al. 2017b).

Chlorophyll decreased sharply during the period of observation at LD-B demonstrating a collapsing *Trichodesmium* bloom (Caffin et al, 2017). Concomitantly, the abundance and taxonomic composition of zooplankton remained homogeneous in the water column. But the abundance of swimmers in the sediment traps decreased by half (Caffin et al, 2017) suggesting an associated reduced zooplankton activity (production, vertical migration) not associated with high mortality. In contrast, the abundance and biomass of zooplankton in the ultra oligotrophic waters of the subtropical Pacific gyre (GY), were

substantially lower than the MA region (W-MA and CE-MA), linked to a far lower primary production mainly concentrated in a deeper chlorophyll maximum, 115–150m depth, in the GY waters (Van Wambeke et al this issue; Moutin et al, this issue).

The taxonomic structure found during OUTPACE (April-May) in the 4 zones (W-MA, CE-MA, BL and GY) showed

a high degree of similarity in term of species richness and abundance distribution across the whole region, however with a moderate difference in subtropical Pacific gyre (GY) based on the relative abundance of a few taxa (see our Fig.8). Copepods were the most abundant group (68 to 86% of total abundance along the transect), showing a slight increase of their contribution from west to east and, in parallel, a slight decrease of gelatinous plankton (mainly dominated by appendicularians) and other holoplankton. The high importance of gelatinous grazers in the W-MA (12-25 %) is consistent

with general attributes of tropical marginal seas, where protistan (e.g., ciliates) and gelatinous grazers account for a high proportion of zooplankton primary consumers (Ceccarelli et al, 2013). Conversely, large copepods are more abundant in areas where 'new' nutrients are introduced (McKinnon et al., 2014). In the GY, the copepod contribution to mesozooplankton was higher than in MA and LD-B (80-86%), mostly in the small size classes (see our Table 3).

There are no data available that have the same spatial coverage of the OUTPACE transect for comparative purposes. At a regional scale, similar results were described by Smeti et al (2015) for the NECTALIS campaigns in July (NECTALIS 1 campaign) and December (NECTALIS 2) in the Coral Sea. They identified 10 common taxa explaining 63% of the similarity in all zones identified in both studies: *Clausocalanus/Paracalanus*, Appendicularians, *Oncaea*, *Oithona*, *Corycaeus*, Ostracods, Chaetognaths, *Lucicutia*, *Mecynocera* and *Acartia*. These 10 taxa should thus constitute the bulk of



the zooplankton assemblage in the western tropical South Pacific (WTSP) and their predominance in all the different zones in our study identified suggests a relative stability in zooplankton composition, despite space and time heterogeneity in environmental conditions. Similarly, in the southern part of the Coral Sea, Rissik et al. (1997), using size and taxonomic information from image analysis of net-collected samples, also showed that the composition of zooplankton was relatively

stable in the different stations studied. Composition was dominated by *Pleuromamma, Acartia,* and *Oncaea,* with a significant contribution of Ostracods and Chaetognaths. Notably, analysis of the tintinnid ciliate community at stations LD-B and LD-C found a similar species richness, abundance distribution and size structure, with only the morphological diversity presenting some differences (Dolan et al. 2016).

      Data on the abundances and biomasses of mesozooplankton in the WTSP (**Table 7**) are scarcer than in the equatorial

waters (Le Borgne et al. 2011) and the eastern subtropical Pacific (Fernández-Álamo & Färber-Lorda, 2006). **Table 7** shows a general consistency between all these data for the tropical area, although variations could be discussed with respect to sampling season, the spatio-temporal variations of regional physical patterns, and the sampling methods used (net gyres, net mesh size, sampling depths, etc.). During OUTPACE, vertical Bongo net tows integrated the 0-200 m water column prohibiting analysis of vertical distributions. However, the small zooplankton (< 1000 μm) was probably concentrated in the

upper 50-70 m, as shown by Smeti et al. (2015) using TAPS and Hydrobios nets, while larger organisms were likely more evenly distributed in the upper 200 m, with maximum values near the DCM. This pattern was consistent with the UVP data obtained during OUTPACE (Unpublished data) where large particles (mainly zooplankton) were more uniformly distributed in the upper 200 m, whereas small size particles (largely dominated by detritus) were concentrated in the upper 50 m.

      On the western side of our transect, in the Coral sea, our data can be compared to those of Smeti et al. (2015) obtained

at different seasons in oceanic waters around New Caledonia (**Table 7**). The stations situated between New Caledonia and the Loyauté Islands registered the highest abundance and biomass values during the cold season, but also the largest variations between stations. Around Mahé, Le Borgne et al (1985) found values ranging from 2.5 to 7 mg DW m$^{-3}$. In contrast, Le Borgne et al. (2011) found slightly lower biomass values than those observed during OUTPACE for oceanographic stations situated nearer to New Caledonia (see **Table 7**). All of these results highlight that the various

mesoscale structures linked to flow disturbance in these oligotrophic bodies of water such as the Coral Sea have a significant effect on the distribution and abundance of zooplankton, imparting substantial heterogeneity, while also being the main seasonal driver of productivity in the region (Menkes et al., 2014; Smeti et al., 2015).On the eastern side of our OUTPACE transect, the few data from the equatorial and sub-tropical waters south-east of Marquesas Islands, between 120-140°WBIOSOPE survey, give comparably low biomass levels (2 to 2.5 mg DW m$^{-3}$; BIOSOPE survey – Table 7). During

the EastroPac cruise at the latitudes 20°N-20°S and longitude 110°W, Longhurst (1976) found abundance values ranging between 100 and 900 indm$^{-3}$, similarly to our observations. He noted that copepods were the dominant taxa, followed by chaetognaths and euphausiids. Between these two ends of the OUTPACE transect, no data were found for comparison with our observations. The obvious increased abundance and biomass in the MA (W-MA and CE-MA) region compared to the



GY region is linked to waters of the Melanesian Archipelago being enriched by contact with multiple islands compared to the ultra-oligotrophic characteristics of the gyre (Rousselet et al. this issue).

There is somehow more information on zooplankton biomass and abundance in the Equatorial Pacific collected during the JGOFS program (Muray et al. 1995 and Le Borgne & Landry (2003). Le Borgne et al. (1999) studied the zonal variability of zooplankton and particle export in April-May 1996 in the equatorial Pacific upwelling between 165° E and 150° W. This parallel transect to OUTPACE showed a general decreasing trend of zooplankton biomass from 14.4 mg DW m$^{-3}$ at the eastern end to 8 mg DW m$^{-3}$ at the western end (Le Borgne et al., 1999, their Fig.3) which was associated with a decrease of Chl-$a$. Almost all studies comparing zooplankton biomass sampled from the equator towards the tropic show as well a strong decrease (Ikeda, 1985, his fig 3B; Dai et al. 2016; White et al.,, 1995; Fernández-Álamo & Färber-Lorda2006; Le Borgne et al, 2003).

Zooplankton diel vertical migration and a secondary bulk of biomass values below 200 m are two major features observed in zooplankton studies with multilayer net tows (such as Hydrobios) in the equatorial and tropical Pacific (Le Borgne et al., 1985; Roman et al;, 1995; Fernández-Álamo & Färber-Lorda2006; Landry et al., 2008; Smeti et al., 2015; Le Borgne et al., 2003). As for our observations in LD-A and LD-B, Smeti et al. (2015) noted that all zooplankton biomass and abundance values were significantly higher during the night than during the day for the larger forms, and attributed this pattern to diel vertical migrations from ADCP-derived zooplankton biomass. The pattern was stronger when surface phytoplankton biomass was higher, which was the case during OUTPACE due to the high contribution of diazotrophs at this season. In the W-MA and CE-MA regions, copepods dominated the mesozooplankton taxonomic structure, but the sampling of teleost eggs and juveniles of euphausiaceae, although certainly under-sampled with our bongo net, do indicate the presence of higher trophic levels in deeper waters in this region (Roger et al. 1994; Bertrand et al., 1999).

In the long duration stations LD-A and LD-B chosen to understand the impact of ephemeral blooms on the ecosystem response and fate of the primary production, the zooplankton population responded with a large production of larval forms. This secondary production increased abundance levels but yielded limited (station LD-A) or no (station LD-A) biomass changes. This zooplankton growth probably induced the high quantities of detritus obtained in the Coral Sea and in the subtropical Pacific gyre.

**4.3 Zooplankton association with diazotrophs during OUTPACE**

The OUTPACE transect was undertaken in a region known for its high N$_2$ fixation (Dupouy et al., 2011) which can contribute 30–50% of new production (Karl et al. 2002). The Melanesian archipelago (New Caledonia, Vanuatu, Fiji Islands; Niué Island, our W-MA and CE-MA regions) is known for its recurrent large *Trichodesmium* blooms during austral summer conditions, which dominates the diazotroph community (Moutin et al., 2005; Bonnet et al., 2015), complemented by high abundances of UCYN-B (Bonnet et al., 2015; Moisander et al., 2010). During OUTPACE, very high values of N$_2$ fixation were registered in most of the W-MA and CE-MA stations, particularly in the upper 25 m, with a slight decrease at SD-9 and SD-10 (Bonnet et al, this issue, their Fig 2.E). N$_2$ fixation was mainly attributed to high concentrations of *Trichodesmium*



and to a lesser extent by UCYN-B (Stenegren et al, 2017; Caffin et al, 2017), and contributed circa 8-12 % of primary production in the W-MA and CE-MA regions (Caffin et al, 2017).

Then N$_2$ fixation rates dropped to much lower values, *circa* one order of magnitude lower than in W-MA and CE-MA regions) in the GY region (Bonnet et al, this issue), with maximum levels occurring deeper in the water column (~50-60 m). The N$_2$ fixation contribution to primary production in the GY region fell to 3 %. The organisms responsible for these fluxes in GY were mainly affiliated with heterotrophic proteobacteria and UCYN-A types), with their maximum of abundance and biomass below 45 m in the euphotic zone (Stenegren et al, 2017). All of these results (community structure and rates) were consistent with results obtained at the western border of the gyre during the BIOSOPE transect (Bonnet et al., 2008; Raimbault and Garcia, 2008; Moutin et al., 2008).

Overall, the horizontal pattern of N$_2$ fixation along the transect showed a good co-variation of N$_2$ fixation with integrated Chl-*a* distribution (Bonnet et al, this issue, their Fig. 2) which means that our zooplankton biomass and abundance fits with the N$_2$ fixation. An abundant diazotroph community is expected to change the structure of the ecosystem, particularly the relative abundance and species composition of grazers and microbial population and the good relationship found between *Trichodesmium* and zooplankton community spatial structuration (first two components of the NDMS analysis on zooplankton taxa) is consistent with this observation.

However, up to now, *Trichodesmium* are thought to be grazed by relatively few mesozooplankton species (Carpenter et al. 1999; Conroy et al., 2017), although new techniques to observe diazotroph in zooplankton gut content may extend this list (Scavotto et al., 2015; Azimuddin et al. 2016; Conroy et al. 2017). Such analysis were not realized during OUTPACE, and we limit our discussion to observed correlation or not of key species distribution with diazotrophs distributions, particularly those among the top 10 species in frequency abundance (Table 4).

The abundance distribution of the Harpacticoid copepods *Macrosetella gracilis and Miracia efferata* and the Poecilostomatoid copepod *Oncaea* showed a positive relationship with *Trichodesmium* abundance (Fig 7A and 7B). The association of *Macrosetella gracilis* with the colonial cyanobacterium *Trichodesmium* has been shown in several studies and is interpreted as a successful way of living within the plankton by using filaments as a physical substrate for juvenile development and/or as a food source, being immune to cyanobacterial toxins harmful to other species of copepods (O'Neil and Roman, 1994; Eberl and Carpenter, 2007). The association between *Oncaea and Trichodesmium* appears possible from our observations as the relationship between their abundances was significant (Fig 7B). A relationship between *Oncaea* and *Trichodesmium* was previously suggested by Dupuis et al (2016) in the Indian Ocean around Madagascar, based on stable isotope data. They interpreted this association as direct consumption of filaments, but also possibly indirect as *Oncaea* species have been shown to have a predominantly omnivorous/detritivorous diet (Atkinson, 1998) being able to use their sharp maxillipeds to catch large prey, such as chaetognaths (Go et al., 1998) and appendicularian houses (Nishibe et al., 2015). Thus, the detritus and aggregates associated with *Trichodesmium*, and even the large trichomes, may benefit this copepod. We found no significant relationship for *Pleuromamma* and *Euchaeta*, despite their association with *Trichodesmium* observed by Azimuddin et al (2016) in the western Pacific, and for Corycaeus.



It is worth noting that we found a positive correlation between pteropods (*Thecosomata*) and *Trichodesmium*, with decreasing abundance of this zooplankton group in GY compared to the other zones. This is interesting because pteropods where in the top ten rank taxa in each zone, representing 1 to 10% of the total zooplankton abundance in the water column and up to 35% of the swimmers in the sediment traps (Caffin et al. 2017). As far as we are aware, a link between *Trichodesmium* and Pteropods (as suggested by our results) has never been established. This association is probably indirect as Pteropods are known as strictly herbivorous filter-feeders with the ability to feed on particles <2µm (Sommer et al, 2002). They have also be found to aggregate at river front limit (Rhone River plume) suggesting that they could be the terminal consumer of the bacterial food chain (Gaudy et al 1996). In this study we did not considered possible association between zooplankton taxa and non- *Trichodesmium* diazotrophs but Hunt et al (2016) from $^{15}N_2$ labelled grazing experiments provided evidence for direct ingestion and assimilation of UCYN-C-derived N by the zooplankton. Recent observations suggested consumption of UCYN-A and UCYN-B by diverse calanoid copepods (Scavotto et al., 2015; Conroy et al., 2017).

From the quantification of the contribution of Diazotroph Derived Nitrogen to Zooplankton $\delta^{15}N$ values, it appeared that the diazotroph contributed up to 67 and 75% to zooplankton biomass in the W-MA and CE-MA regions respectively, but strongly decreased to an average of 22% in the SG region and down to 7% occurring in the eastern station. The highest values of ZDDN were comparable to the highest value (73%) observed during the VAHINE mesocosm experiment in the oligotrophic New Caledonia lagoon (Hunt et al. 2016), in one experiment stimulating UCYN-C bloom accompanied to *Trichodesmium* spp., DDA (*Richelia* associated with the diatom *Rhizosolenia* and the diatom *Hemiaulus* at lower concentrations

## 4.4. Fluxes associated with zooplankton

The weight specific rates of ingestion and NH4 and PO4 excretion (0.5-0.6 d$^{-1}$, 0.10-0.14 d$^{-1}$ and 0.09-0.11 d$^{-1}$ respectively) calculated using the allometric relationships of Ikeda (1985), based on individual dry weight estimates of the different zooplankton taxa, and on temperature and chlorophyll levels recorded during OUTPACE, were in the range of literature values for metazooplankton and copepods in the inter-tropical zone (Ikeda, 1985; Dam et al, 1995; Mauchline, 1998; Hernandez-Leon et al, 2008; McKinnon et al, 2015), thus validating this approach.

The estimated ingestion and metabolic rates allowed us to estimate that the top-down (through grazing) and bottom-up (through excretion of N and P) impact of zooplankton on phytoplankton were potentially high in the OUTPACE zone. The top-down impact was particularly high in the MA zone - W-MA and CE-MA - (up to 234% of the daily primary production, mean = 102±61%) in agreement with Moutin et al. (this issue) who pointed out that a strong top-down pressure by zooplankton grazing in the MA maintains low level pigment concentration in a quasi-steady state for many months. Zooplankton grazing represented a daily removal of 3.6 to 21% of the phytoplankton stock and of 16 to 234% of the primary production, which was high compared to those by Dam et al (1995) for the equator zone in the central Pacific: 1 to 9% of the phytoplankton stock and 2 to 13% of the primary production.





Such strong grazing impact on primary producers emphasizes the role of zooplankton in the sink of athmospheric CO2atmospheric $CO_2$ as also underlined by Moutin et al (this issue). In addition, through quantification of the major biogeochemical fluxes these authors suggest that mesozooplankton diel vertical migration plays a dominant role in the transfer of carbon from the upper surface to deeper water in the MA zone. Our observations also clearly support diel vertical migration of zooplankton in the MA zone as zooplankton biomass clearly increased in night samples compared to day samples in LD-A and LD-B, mainly due to the size fraction larger than 2000 μm (euphausids, large copepods, etc.).

From our results, we can also estimate that the top down impact of zooplankton on $N_2$fixators must be important. Indeed, Caffin et al (this issue) estimated that $N_2$fixation contributed circa 8-12 % of primary production in the MA region and 3 % in the SPG water and sustained nearly all new primary production at all stations. As zooplankton grazing remove 16 to 234% of the total primary production daily, we can estimate that 2 to 100% of $N_2$fixing organisms should be removed daily. This strong impact on diazotrops is also supported by the fact that as shown by isotopic approach, the diazotroph contributed up to 67 and 75% to zooplankton biomass in the W-MA and CE-MA regions respectively.

Regeneration of nutrients by zooplankton excretion was high suggesting high contribution to regenerated production particularly in terms of nitrogen. Daily $NH_4$ excretion represented 15 to 165 % of phytoplankton needs for N whereas PO4 excretion accounted only for 3 to 34% of P needs. The estimates for NH4 regeneration are in the upper range of literature data summarized by Hernandez-Leon et al (2008) and Le Borgne (1986) for different areas of the world ocean: from 2 to 100% and up to 82% in the Narraganset bay. They are higher than values reported for the central tropical Pacific (up to 17%; Dam et al, 1995; Zhang et al 1995), the equatorial Pacific (31–36%; Gaudy et al.; 2003) and for the North Pacific central gyre (40–50%; Eppley et al., 1973), but rather close to the values recorded in the Atlantic Ocean between 50°N and 30°S (31–100%; Isla et al., 2004). Our estimates of the contribution of phosphorous excretion to phytoplankton requirements (3 to 34% d-1) are also in the range of the literature values reviewed by Leborgne (1985), from 2 % in the NW Mediterranean to 200 % in Narraganset bay.

During OUTPACE, there was no clear spatial trend in top - down (grazing) vs bottom-up (N and P regeneration) zooplankton impacts on phytoplankton despite both processes appeared important in all sites. However, Bock et al (this issue) showed that microbial communities along the transect were predominantly regulated by bottom-up control processes at mesotrophic stations, with top-down processes becoming increasingly important at oligotrophic stations. In particular, they showed a clear reduction in top-down control of heterotrophic nanoflagellates in oligotrophic stations, largely in agreement with previous studies (Gasol et al., 2002). The strong decrease of tintinnid abundance in GY (LD-C) compared to BL (LD-B) evidenced by Dolan et al (2016) also argue in favor of a decreased top-down control of microbial communities in more oligotrophic regions.

As a whole, the OUTPACE survey conducted during the main blooming season of the South western tropical Pacific demonstrated that this key production for the pelagic food chain was dependent of the conjunction of physical mesoscale pattern, availability of micro and macronutrient, and growing capabilities of autotrophic diazotrophs. The contribution of this primary production to mesozooplankton is probably an essential trophic link in the tuna marine food web.





**Acknowledgements**

This is a contribution of the OUTPACE (Oligotrophy from Ultra-oligoTrophy PACific Experiment) project funded by the French research MZAtional agency (ANR-14-CE01-0007-01), the LEFE-CyBER program (CNRS-INSU), the GOPS program (IRD) and the CNES (BC T23, ZBC 4500048836) and by European FEDER Fund under project 1166-39417 The

5  OUTPACE cruise (DOI: http://dx.doi.org/10.17600/15000900) was managed by T. Moutin and S. Bonnet from the MIO (Mediterranean Institute of Oceanography). The authors thank to the crew of the R/V L´Atalante for outstanding shipboard operation. This work was supported by the Comisión Nacional de Investigación Científicas y Tecnológicas (CONICYT) through Grants FONDECYT No 1130511 and 1150891 and Instituto Milenio de Oceanografía (IMO) Grant IC120019. Valdés and Donoso were funded by CONICYT Scholarship. The authors thank France Van Wambeke, Cécile Dupouy and

10  Marcus Stenegren and for sharing their data on primary production chlorophyll and diazotroph abundance (respectively), that helped for interpretation and comparison with our results



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





**Table 1** Mean values of salinity, temperature, total Chl-*a* and MLD depth found at the stations for the four clusters defined in the PCA analysis on environmental variables (see Fig.1) and for the 3 long duration stations. W-MA = Western Melanesian archipelago, CE-MA= Central and Eastern Melanesian archipelago, BL = station B (blooming conditions) and GY = subtropical gyre

| | W-MA | CE-MA | BL | GY | LD-A | LD-B | LD-C |
|---|---|---|---|---|---|---|---|
| Salinity | 35.27 ± 0.41 | 35.58 ± 0.03 | 36.31 | 35.86 ± 0.30 | 35.43 ± 0.07 | 36.31 ± 0.48 | 36.19 ± 0.57 |
| Temperature | 24.59 ± 0.78 | 23.95 ± 0.65 | 25.38 | 25.36 ± 0.36 | 25.38 ± 0.63 | 25.38 ± 0.45 | 24.96 ± 0.68 |
| total chla | 0.42 ± 0.06 | 0.38 ± 0.09 | 0.48 | 0.19 ± 0.04 | 0.42 ± 0.07 | 0.48 ± 0.10 | 0.23 ± 0.02 |
| %chla | 70.64 ± 6.60 | 56.72 ± 2.45 | 67.07 | 55.63 ± 3.77 | 61.80 ± 3.51 | 67.07 ± 1.47 | 59.64 ± 2.34 |
| MLD | 14.69 ± 4.46 | 15.67 ± 5.34 | 26.75 | 34.25 ± 5.63 | 16.75 ± 5.56 | 26.75 ± 6.13 | 28.75 ± 9.52 |




**Table 2.** Mean and SD values of zooplankton abundances, percentage of taxonomic groups and total copepod demographic parameters at the stations for the four clusters defined in the PCA analysis on environmental variables (see Fig.1) and for the 3 long duration stations. W-MA = Western Melanesian archipelago, CE-MA= Central and Eastern Melanesian archipelago, BL = station B (blooming conditions) and GY = subtropical gyre

| | W-MA | CE-MA | BL | GY | LD-A | LD-B | LD-C |
|---|---|---|---|---|---|---|---|
| **Zooplankton zooscan** | | | | | | | |
| total > 300 µm ECD (ind m-3) | 718 ± 226 | 527 ± 120 | 678 | 250 ± 52 | 687 ± 233 | 678 ± 144 | 290 ± 72 |
| **Zooplankton microscope** | | | | | | | |
| > 300 µm ECD (ind m$^{-3}$) | 634 ± 169 | 724 ± 208 | 648 | 357 ± 91 | 684 ± 192 | 648 ± 103 | 404 ± 26 |
| Total (ind m-3) | 1179 ± 370 | 1234 ± 358 | 1145 | 655 ± 213 | 1198 ± 520 | 1145 ± 175 | 784 ± 59 |
| % Copepods | 73.1 ± 4.6 | 76.4 ± 3.1 | 77.4 | 82.3 ± 3.6 | 68.4 ± 11.6 | 77.4 ± 0.8 | 85.9 ± 1.9 |
| % Gelatinous | 16.3 ± 3.7 | 13.5 ± 2.5 | 12.6 | 11.3 ± 5.6 | 21.0 ± 10.8 | 12.6 ± 1.8 | 7.5 ± 1.4 |
| % Chaetognaths | 2.2 ± 1.6 | 2.5 ± 0.7 | 2.0 | 0.6 ± 0.1 | 3.3 ± 0.6 | 2.0 ± 0.9 | 0.8 ± 0.2 |
| % Other holoplankton | 7.8 ± 3.5 | 6.7 ± 1.3 | 7.1 | 4.9 ± 2.4 | 6.4 ± 2.1 | 7.1 ± 2.2 | 5.3 ± 1.4 |
| % Meroplankton | 0.6 ± 0.3 | 0.9 ± 0.6 | 0.8 | 0.9 ± 0.5 | 0.9 ± 0.2 | 0.8 ± 0.6 | 0.5 ± 0.2 |
| **Copepods** | | | | | | | |
| Total (ind m$^{-3}$) | 862 ± 17 | 943 ± 11 | 887 | 539 ± 8 | 834 ± 427 | 887 ± 144 | 659 ± 55 |
| %nauplii | 13.3 ± 3.1 | 15.9 ± 9.6 | 14.4 | 11.1 ± 7.1 | 13.6 ± 5.5 | 14.4 ± 4.5 | 12.4 ± 4.3 |
| % copepodites | 68.3 ± 6.1 | 60.7 ± 10.7 | 67.0 | 74.3 ± 7.0 | 61.7 ± 2.8 | 67.0 ± 5.5 | 70.6 ± 4.7 |
| % adults | 18.4 ± 6.8 | 23.3 ± 4.6 | 18.5 | 14.7 ± 2.2 | 24.7 ± 4.9 | 18.5 ± 2.7 | 17.0 ± 3.8 |
| sex ratio ( % females/adults) | 70.4 ± 8.3 | 79.7 ± 7.9 | 78.5 | 78.8 ± 5.9 | 62.6 ± 41.9 | 78.5 ± 7.9 | 78.9 ± 4.0 |





**Table 3**: Mean and SD values of zooplankton biomasses values, percentage of total biomass for the different size fractions, and phytoplankton biomass estimated from Chla, at the stations for the four clusters defined in the PCA analysis on environmental variables (see Fig.1) and for the 3 long duration stations. W-MA = Western Melanesian archipelago, CE-MA= Central and Eastern Melanesian archipelago, BL = station B (blooming conditions) and GY = subtropical gyre

| | W-MA | | | CE-MA | | | BL | GY | | | LD-A | | | LD-B | | | LD-C | | |
|---|---|---|---|---|---|---|---|---|---|---|---|---|---|---|---|---|---|---|---|
| **Biomass mg DW m⁻³)** | | | | | | | | | | | | | | | | | | | |
| Phytoplankton | 0.0 | ± | 0.0 | 0.0 | ± | 0.0 | 0.0 | 0.0 | ± | 0.0 | 42.5 | ± | 6.7 | 48.2 | ± | 9.8 | 23.2 | ± | 1.9 |
| Zooplankton (weighed) | 12.2 | ± | 5.5 | 6.5 | ± | 4.0 | 10.6 | 2.5 | ± | 0.2 | 12.4 | ± | 2.1 | 10.6 | ± | 1.7 | 2.7 | ± | 0.4 |
| Zooplankton(micro + zooscan) | 5.7 | ± | 1.6 | 5.6 | ± | 1.9 | 8.9 | 2.0 | ± | 0.7 | 7.9 | ± | 2.9 | 8.9 | ± | 4.1 | 2.8 | ± | 1.3 |
| Zooplankton <300 µm (micros) | 0.34 | ± | 0.14 | 0.31 | ± | 0.10 | 0.30 | 0.18 | ± | 0.08 | 0.32 | ± | 0.22 | 0.30 | ± | 0.05 | 0.23 | ± | 0.02 |
| Zooplankton >300 µm (Zooscan) | 5.3 | ± | 1.4 | 5.2 | ± | 1.8 | 8.6 | 1.8 | ± | 0.7 | 7.6 | ± | 2.7 | 8.6 | ± | 4.1 | 2.6 | ± | 1.2 |
| detritus (zooscan) | 4.7 | ± | 1.6 | 2.9 | ± | 2.5 | 3.9 | 1.0 | ± | 0.0 | 7.0 | ± | 2.1 | 3.9 | ± | 1.2 | 1.0 | ± | 0.7 |
| zooplankton + detritus | 10.4 | ± | 3.1 | 8.5 | ± | 4.3 | 12.8 | 3.0 | ± | 0.7 | 14.9 | ± | 2.6 | 12.8 | ± | 1.0 | 3.8 | ± | 1.0 |
| **% detritus** | 44.9 | ± | 5.2 | 30.4 | ± | 10.9 | 30.7 | 35.2 | ± | 8.6 | 46.8 | ± | 13.4 | 30.7 | ± | 5.2 | 26.5 | ± | 5.8 |
| **% Zooplankton biomass** | | | | | | | | | | | | | | | | | | | |
| <300 µm (micro) | 6.3 | ± | 3.5 | 6.0 | ± | 1.8 | 3.4 | 9.1 | ± | 2.8 | 4.0 | ± | 2.7 | 3.4 | ± | 1.1 | 8.3 | ± | 2.2 |
| 300-500 µm (zooscan) | 25.4 | ± | 5.1 | 23.2 | ± | 3.0 | 17.4 | 33.1 | ± | 5.2 | 19.9 | ± | 5.1 | 17.4 | ± | 4.2 | 28.4 | ± | 7.9 |
| 500-1000 µm (Zooscan) | 35.7 | ± | 6.4 | 33.1 | ± | 2.5 | 33.0 | 26.4 | ± | 0.9 | 32.3 | ± | 9.2 | 33.0 | ± | 4.9 | 27.2 | ± | 3.7 |
| 1000-2000 µm (Zooscan) | 18.0 | ± | 7.0 | 21.1 | ± | 2.6 | 21.9 | 17.9 | ± | 3.8 | 23.5 | ± | 3.3 | 21.9 | ± | 4.1 | 20.3 | ± | 7.1 |
| >2000 µm (Zooscan) | 14.7 | ± | 7.2 | 16.6 | ± | 6.3 | 24.3 | 13.5 | ± | 4.7 | 20.4 | ± | 10.9 | 24.3 | ± | 10.0 | 15.9 | ± | 7.0 |




**Table 4:** Top 10 species in frequency abundance for the 4 regions (W-MA = Western Melanesian archipelago, CE-MA= Central and Eastern Melanesian archipelago, BL = station B ,blooming conditions  and GY = subtropical gyre).

| Rank | W-MA | CE-MA | BL | GY |
|---|---|---|---|---|
| 1 | *Clauso/Paracalanus* | *Clauso/Paracalanus* | *Oncaea* | *Clauso/Paracalanus* |
| 2 | *Appendicularia* | *Oithona* | *Clauso/Paracalanus* | *Oithona* |
| 3 | *Oncaea* | *Oncaea* | *Oithona* | *Corycaeus* |
| 4 | *Oithona* | *Appendicularia* | *Appendicularia* | *Appendicularia* |
| 5 | *Nauplii* | *Nauplii* | *Nauplii* | *Oncaea* |
| 6 | *Corycaeus* | *Corycaeus* | *Microsetella* | *Nauplii* |
| 7 | *Thecosomata* | *Microsetella* | *Ostracoda* | *Microsetella* |
| 8 | *Microsetella* | *Thecosomata* | *Corycaeus* | *Thecosomata* |
| 9 | *Calocalanus* | *Ostracoda* | *Thecosomata* | *Calocalanus* |
| 10 | *Chaetognatha* | *Chaetognatha* | *Chaetognatha* | *Mecynocera clausi* |





**Table 5:** Mean values per region of taxonomic diversity (H' = Shannon index) and taxonomic richness (nb taxa per sample) calculated for total zooplankton and copepod communities. W-MA = Western Melanesian archipelago, CE-MA= Central and Eastern Melanesian archipelago, BL = station B, blooming conditions  and GY = subtropical gyre.

| | W-MA | | | CE-MA | | | BL | GY | | | | LD-A | | | LD-B | | | LD-C | | |
|---|---|---|---|---|---|---|---|---|---|---|---|---|---|---|---|---|---|---|---|---|
| H' zooplankton | 3.54 | ± | 0.07 | 3.66 | ± | 0.09 | 3.67 | 3.40 | ± | 0.10 | | 3.50 | ± | 0.04 | 3.67 | ± | 0.11 | 3.40 | ± | 0.04 |
| nb taxa zooplankton | 33.00 | ± | 2.94 | 32.56 | ± | 4.75 | 34.00 | 31.33 | ± | 1.15 | | 31.00 | ± | 8.12 | 34.00 | ± | 4.97 | 32.00 | ± | 3.42 |
| H' copepods | 3.08 | ± | 0.08 | 3.14 | ± | 0.09 | 3.13 | 2.91 | ± | 0.02 | | 3.06 | ± | 0.18 | 3.13 | ± | 0.08 | 2.92 | ± | 0.06 |
| nb taxa copepods | 21.81 | ± | 1.68 | 22.78 | ± | 3.70 | 22.75 | 20.33 | ± | 1.15 | | 21.25 | ± | 5.62 | 22.75 | ± | 3.10 | 21.00 | ± | 2.94 |





**Table 6.** Zooplankton weight specific ingestion and excretion rates in relation to phytoplankton carbon stocks and primary production values, and zooplankton grazing and nutrient regeneration impacts. W-MA = Western Melanesian archipelago, CE-MA= Central and Eastern Melanesian archipelago, BL = station B, blooming conditions  and GY = subtropical gyre.

| | W-MA | | | CE-MA | | | BL | GY | | | LD-A | | | LD-B | | | LD-C | | |
|---|---|---|---|---|---|---|---|---|---|---|---|---|---|---|---|---|---|---|---|
| **weight specific rates** | | | | | | | | | | | | | | | | | | | |
| Ingestion (d$^{-1}$) | 0.62 | ± | 0.02 | 0.59 | ± | 0.02 | 0.64 | 0.57 | ± | 0.01 | 0.64 | ± | 0.02 | 0.64 | ± | 0.03 | 0.56 | ± | 0.02 |
| NH$_4$ Excretion (d$^{-1}$) | 0.124 | ± | 0.012 | 0.115 | ± | 0.009 | 0.107 | 0.139 | ± | 0.021 | 0.122 | ± | 0.013 | 0.107 | ± | 0.008 | 0.115 | ± | 0.007 |
| PO$_4$ Excretion (d$^{-1}$) | 0.101 | ± | 0.009 | 0.096 | ± | 0.007 | 0.088 | 0.111 | ± | 0.014 | 0.100 | ± | 0.009 | 0.088 | ± | 0.006 | 0.095 | ± | 0.004 |
| **Grazing impact on phytoplankton** | | | | | | | | | | | | | | | | | | | |
| Phytoplankton (mC m-2) | 2104 | ± | 286 | 1877 | ± | 432 | 2412 | 887 | ± | 89 | 2124 | ± | 333 | 2412 | ± | 1 | 973 | ± | 1 |
| Primary production (mgC m$^{-2}$d$^{-1}$) | 482 | ± | 134 | 263 | ± | 162 | 472 | 87 | ± | 27 | 656 | | | 472 | | | 116 | | |
| ZCD ( mgC m$^{-2}$ d$^{-1}$) | 184 | ± | 84 | 212 | ± | 81 | 332 | 66 | ± | 18 | 285 | ± | 109 | 332 | ± | 3 | 81 | ± | 3 |
| % phytoplankton stock d$^{-1}$ | 9.1 | ± | 4.5 | 11.3 | ± | 3.9 | 14.7 | 7.3 | ± | 1.4 | 13.8 | ± | 5.7 | 14.7 | ± | 5.9 | 8.4 | ± | 1.7 |
| % primary production d$^{-1}$ | 39.4 | ± | 16.9 | 102.6 | ± | 61.6 | 70.4 | 78.6 | ± | 28.8 | 43.5 | ± | 16.6 | 70.4 | ± | 11.6 | 70.4 | ± | 12.0 |
| **NH$_4$ excretion impact on phytoplankton** | | | | | | | | | | | | | | | | | | | |
| phytoplankton needs (mgN m$^{-2}$d$^{-1}$) | 6.08 | ± | 1.69 | 3.32 | ± | 2.04 | 5.96 | 1.10 | ± | 0.34 | 8.28 | ± | 0.00 | 5.96 | ± | 3.00 | 1.46 | ± | 3.00 |
| Regeneration (mg N-NH$_4$ m$^{-2}$ d$^{-1}$) | 1.75 | ± | 0.71 | 2.01 | ± | 0.66 | 2.75 | 0.77 | ± | 0.16 | 2.63 | ± | 0.94 | 2.75 | ± | 3.00 | 0.82 | ± | 3.00 |
| % N demand d$^{-1}$ | 29.7 | ± | 11.5 | 77.2 | ± | 43.8 | 46.2 | 75.4 | ± | 33.4 | 31.8 | ± | 11.3 | 46.2 | ± | 8.9 | 56.3 | ± | 10.8 |
| **PO$_4$ exretion impact on phytoplankton** | | | | | | | | | | | | | | | | | | | |
| phytoplankton needs (mgP m$^{-2}$d$^{-1}$) | 0.38 | ± | 0.11 | 0.21 | ± | 0.13 | 0.37 | 0.07 | ± | 0.02 | 0.52 | ± | 0.00 | 0.37 | ± | 3.00 | 0.09 | ± | 3.00 |
| Regeneration (mg P-PO$_4$ m$^{-2}$ d$^{-1}$) | 0.02 | ± | 0.01 | 0.03 | ± | 0.01 | 0.03 | 0.01 | ± | 0.00 | 0.03 | ± | 0.01 | 0.03 | ± | 3.00 | 0.01 | ± | 3.00 |
| % P demand d$^{-1}$ | 5.9 | ± | 2.3 | 15.6 | ± | 9.2 | 9.2 | 14.5 | ± | 6.2 | 6.3 | ± | 2.3 | 9.2 | ± | 1.7 | 11.2 | ± | 2.0 |



**Table 7**. Average zooplankton abundance and biomass values from different region of the western and central tropical South Pacific around the 20° parallel south.

| Campain | Region | Lat. | Long. | Abundances (ind m⁻³) | Biomasses (mg DW m⁻³) | Reference |
|---|---|---|---|---|---|---|
| FLUPAC | Equator | 0° | 180° | - | 14-18 | Le Borgne et al (2003) |
| Hydrobios | | 8°S | 180° | - | 5 | Le Borgne et al (1990) |
| BIOSOPE Bongot net, 200 µm | Marquisean islands | 8.4°S | 141°W | - | 15-25 | Carlotti (unpublished data) |
| OUTPACE Bongot net, 120µm | Coral Sea Feb-April 2015 | 17-22°S | 160-170°E | 800-1600 | 4-7.5 | Present paper |
| 19 oceanographic stations in New Caledonia. | Coral Sea | 17-22°S | 160-170°E | - | 2-3 | Le Borgne et al. (2011) |
| NECTALIS 1 HydrobiosMultiNet 200 µm | Coral Sea Cool seasonJuly 2011 | 17-22°S | 160-170°E | 200-400 | 2.5-6.9 | Smeti et al (2015) |
| NECTALIS 2 HydrobiosMultiNet 200 µm | Coral Sea Hot season December 2011 | 17-22°S | 160-170°E | 150-250 | 2.0-2.8 | Smeti et al (2015) |
| BIOSOPE Bongot net, 200 µm | SPSG | 20°S | 130-120°W | - | 2-2.5 | Carlotti (unpublished data) |
| OUTPACE Bongot net, 120 µm | GY | 20°S | 160-165°W | 450-870 | 1.2-2.8 | Present paper |
| PROCAL WP-2 net , 200 µm | Mahé | 21,5°S | 169°E | - | 2.5-7 | Le Borgne et al (1985) |



**Figures caption**

Fig. 1. Transect of the OUTPACE cruise superimposed on quasi-Lagrangian weighted mean chl a of the WTSP during OUTPACE (see details in Moutin et al. 2017) with the two types of stations, short duration stations 1 to 15 (**x**) and long duration stations A, B and C (+). Along the transect, zooplankton samples were collected once at each short duration station, whereas day-night sampling were performed each day at three strategic long duration stations. Longitude is expressed as °E.

Fig. 2. PCA on environmental variables: Mixed-layer Depth (MLD); total Chl-*a* concentration% Chl-*a* (ratio Chl-*a*/ Chl-*a*+Phae), temperature, salinity averaged on the upper 0-200m of the water column. Plots of the 15 stations (A) and variables (B) on the first factorial plan. The green circles delimit the clusters defined at a distance of 3.3: W-MA = Western Melanesian archipelago, CE-MA= Central and Eastern Melanesian archipelago, BL = station B ,blooming conditions  and GY = subtropical gyre.

Fig. 3. Zooplankton abundance along the OUTPACE West-East transect. (A) Abundance (ind m$^{-3}$) of small (ECD <300μm) and large (ECD >300μm) zooplankton determined by microscope counting (vertical bars) and of large zooplankton (ECD >300μm) determined by Zooscan (dark line). Variations of Chl-*a* concentrations (green line). (B) Percentage of abundance for the different zooplankton groups from binocular counting. SD-01 to SD-15: short duration stations. LD-A, LD-B and LD-C: long duration stations (average value and standard deviation over the 5 days sampling).

Fig. 4. Zooplankton biomasses along the OUPACE West-East transect. (A) Cumulated zooplankton biomass and detritus sampled from the bongo net along the transect. Red line - values of total dry weight determined by weighting at each station. Green line - water column chlorophyll. (B) Percentage of biomass for the different size fractions and detritus. Zooplankton biomass fraction < 300 μm was determined from microscopic counting. Zooplankton biomass fractions > 300 μm (4 fractions) and detritus biomass were estimated from Zooscan biovolumes. SD-01 to 15: short duration stations. LD-A, LD-B and LD-C: long duration stations (average value and standard deviation over the 5 days sampling).

Fig. 5. Rank frequency distributions of the total zooplankton community (log of abundance frequency vs. log of rank) for the four regions. In full dark blue circle: W-MA = Western Melanesian archipelago; In full red circle : CE-MA= Central and Eastern Melanesian archipelago; In green triangle: BL = Blooming conditions - station LD-B-; In yellow triangle GY = subtropical gyre).

Fig 6**.** Mean percentage of abundance of the main zooplankton groups or taxa in the five size classes (<300μm; 300-500 μm; 500-1000 μm; 1000-2000 μm; >2000 μm) and for each of the 4 regions: W-MA = Western Melanesian archipelago, CE-MA= Central and Eastern Melanesian archipelago, BL = Blooming conditions - station LD-B-and GY = subtropical gyre.

Fig 7. Spatial variation of abundance of copepod taxa known to have trophic relationships with *Trichodesmium*: *Harpacticoids* (Top panel : cumulated abundance of 3 species) and *Poecilostomatoids* (<middle panel : cumulated abundance of 2 species) and *Calanoidae* (bottom panel: cumulated abundance of 2 species)  comparison with variation of *Trichodesmium*abundance (from Stenegren et al, this issue).

Fig 8. NDMS of the main zooplankton taxa (>0.1% abundance). A: Plot of the stations with different colors between the regions identified with the environmental clustering (4 regions: W-MA = Western Melanesian archipelago, CE-MA= Central and Eastern Melanesian archipelago, BL = Blooming conditions - station LD-B-; and GY = subtropical gyre..). B: Plot of the taxa.

Fig 9. Temporal variation of zooplankton abundance and biomass over 5 days at each of the three long duration stations (LD-A, LD-B and LD-C from left to right). Top panel (A to C): Percentage of abundance for the different zooplankton groups from binocular counting realized for first and last day and night net tows. Bottom panel (C to E): Percentage of cumulated zooplankton biomass and detritus sampled from the bongo net. Zooplankton biomass fraction below 300 μm is determined from microscopic counting. Zooplankton biomass fractions above 300 μm (4 fractions) and detritus biomass are estimated from Zooscan biovolumes.

Fig 10. Comparison of zooplankton abundance and percentage of main taxa: organisms in the water column (0-200 m) (Top panel A to C) and "swimmers" in the sediment trap at 150 m deep (Bottom panel D to E) at each of the three long duration stations (LD-A, LD-B and LD-C from left to right). Sediment trap data at day 5 cannot be considered for the analysis (see Caffin et al, this issue)

Fig. 11. A) Biomass weighted zooplankton and POM (5m depth) nitrogen isotope ratios (δ15N); B) Average percent contribution of diazotroph derived nitrogen (DDN) to zooplankton biomass (ZDDN). W-MA = Western Melanesian archipelago, CE-MA= Central and Eastern Melanesian archipelago, and GY = subtropical gyre.





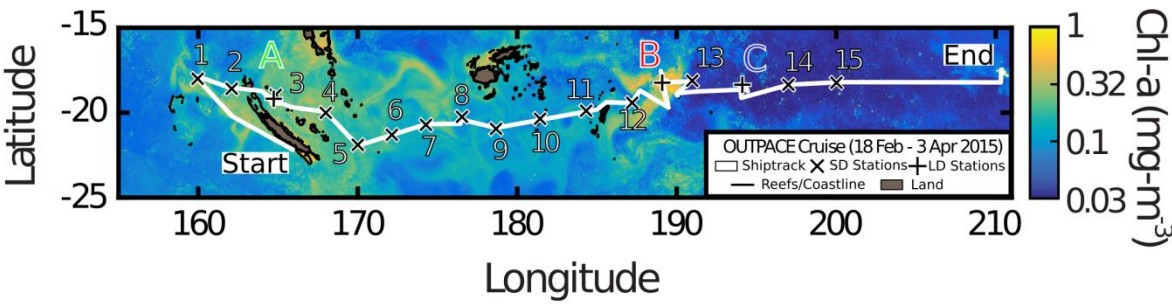

**Figure 1**



**Figure 2.A,B**





**Figure 3.A,B**





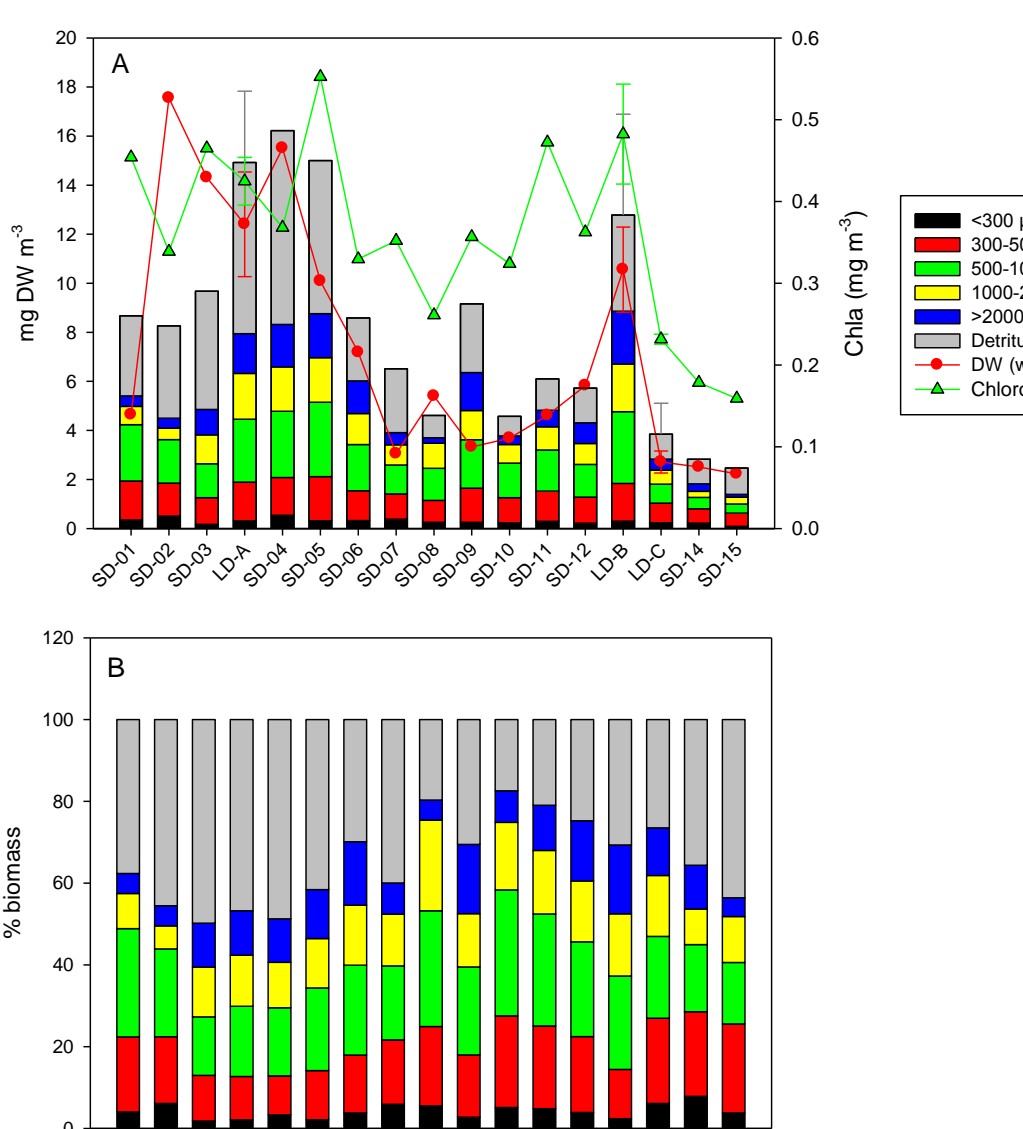

**Figure 4 A,B**





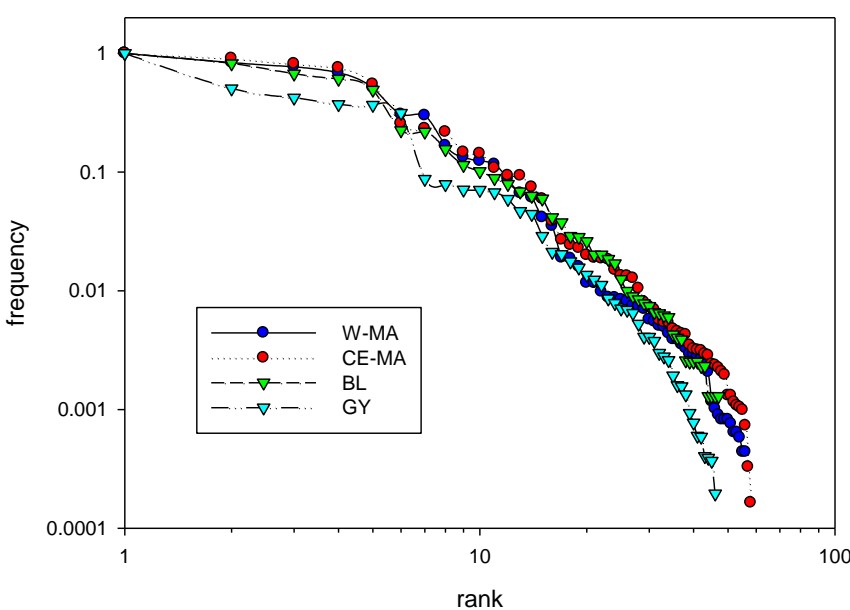

**Figure 5**





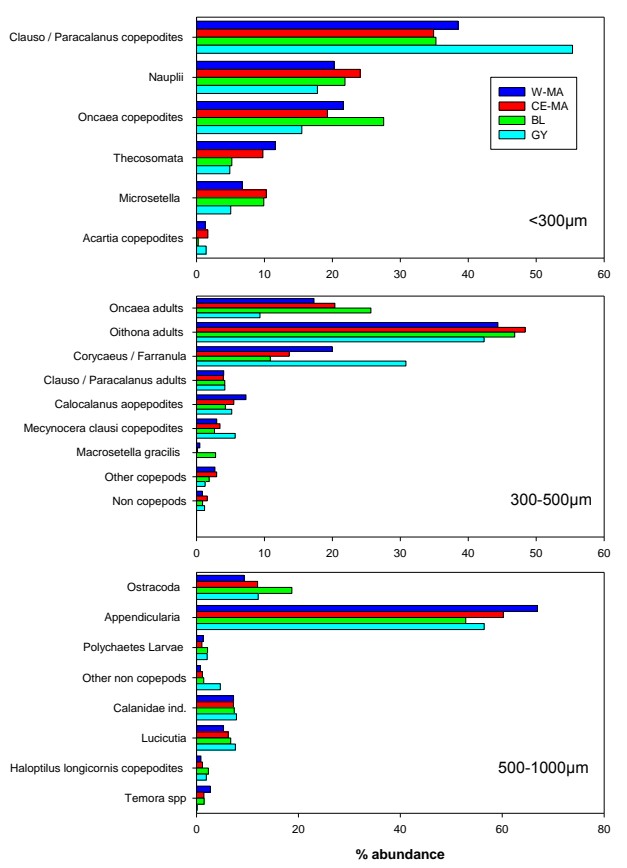

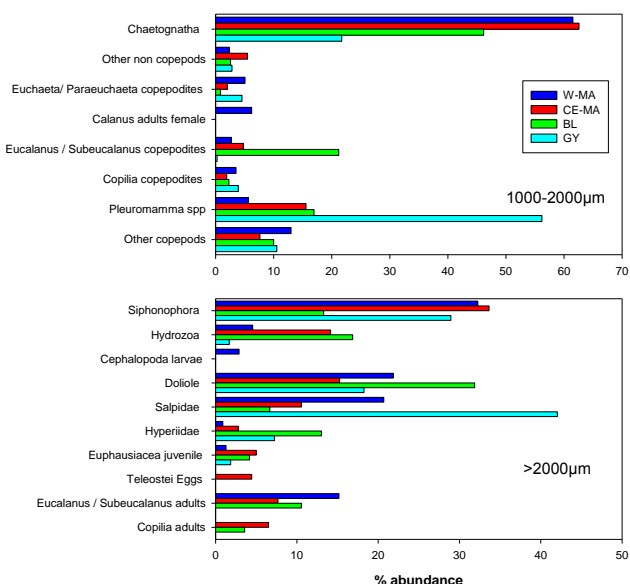

**Figure 6**





**Figure 7 A, B, C**



**A**

**B**

**Figure 8 A, B**





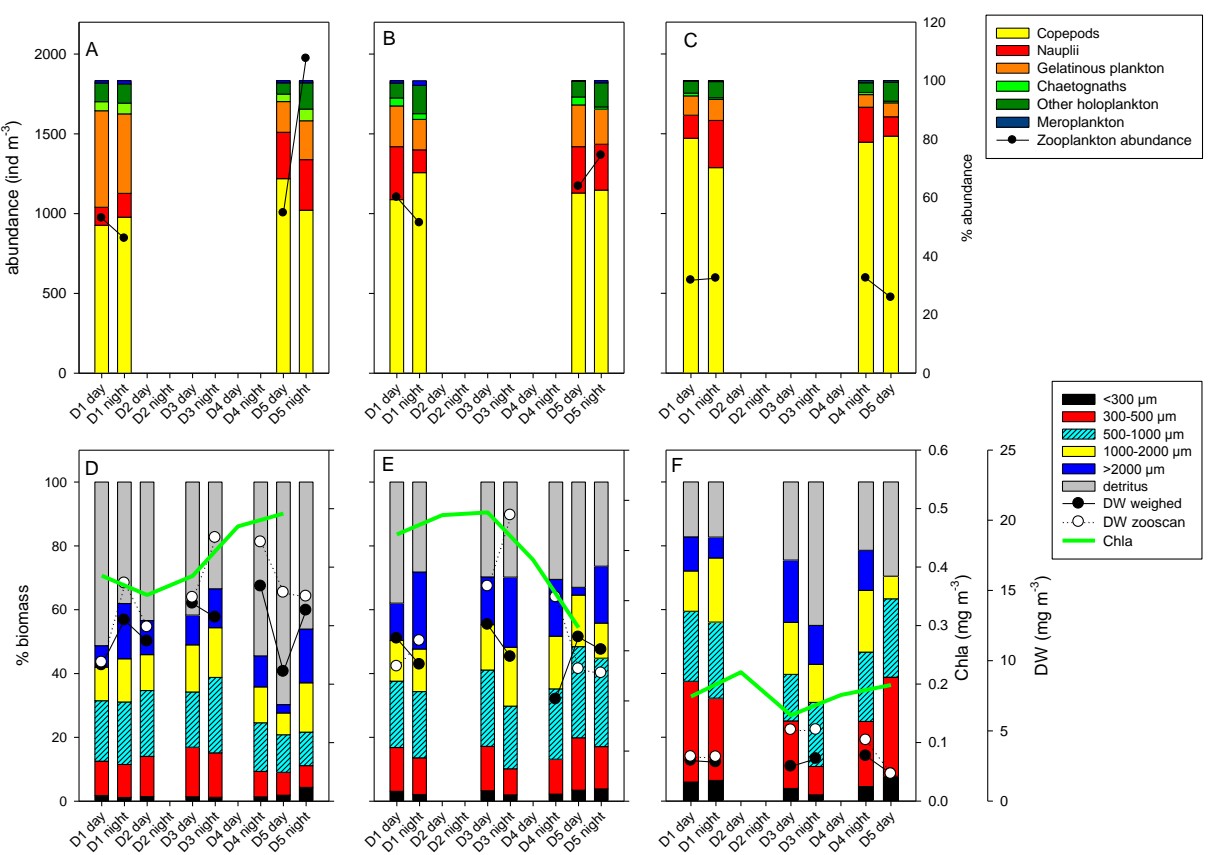

5   **Figure 9 A, B, C, D, E, F**





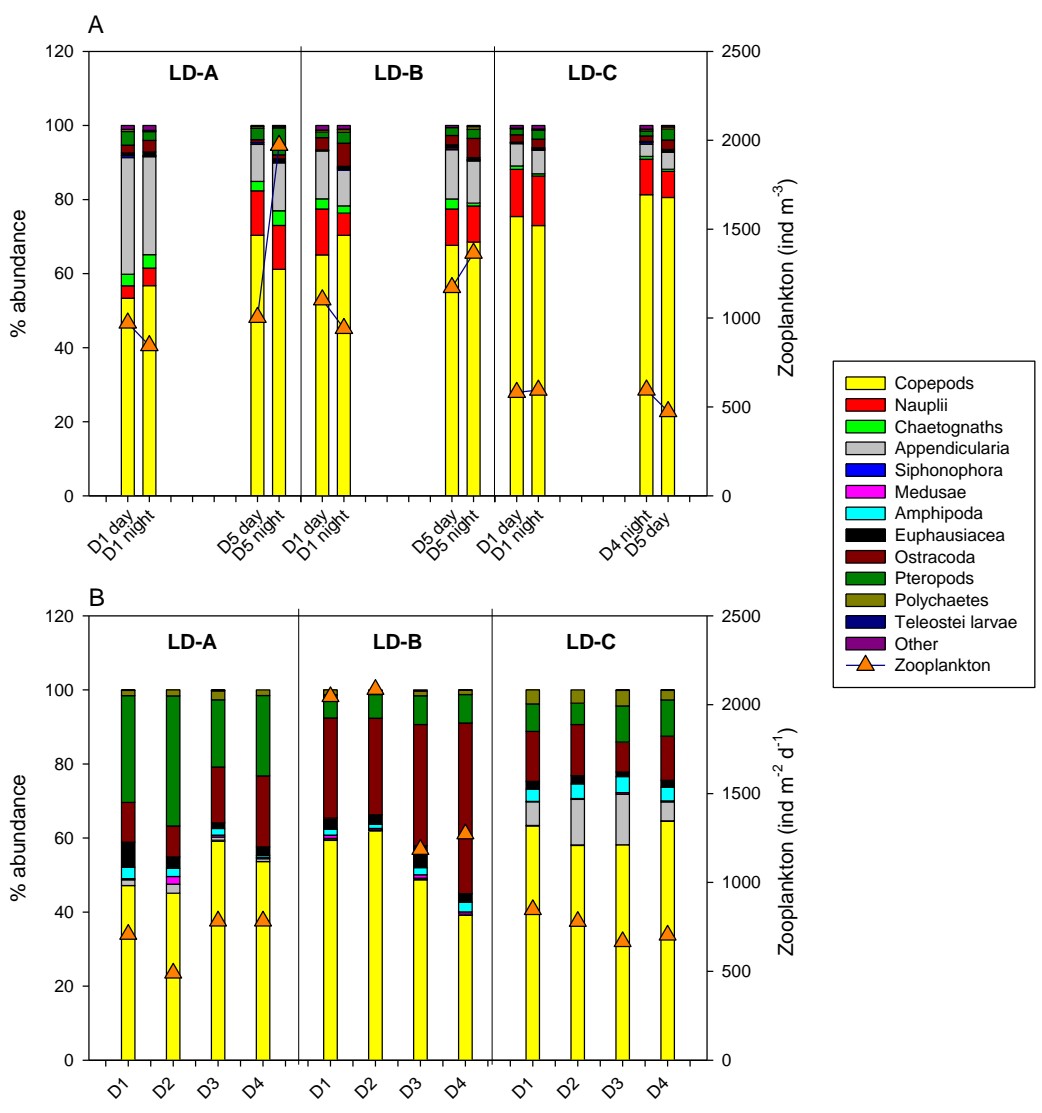

**Figure 10 A, B**




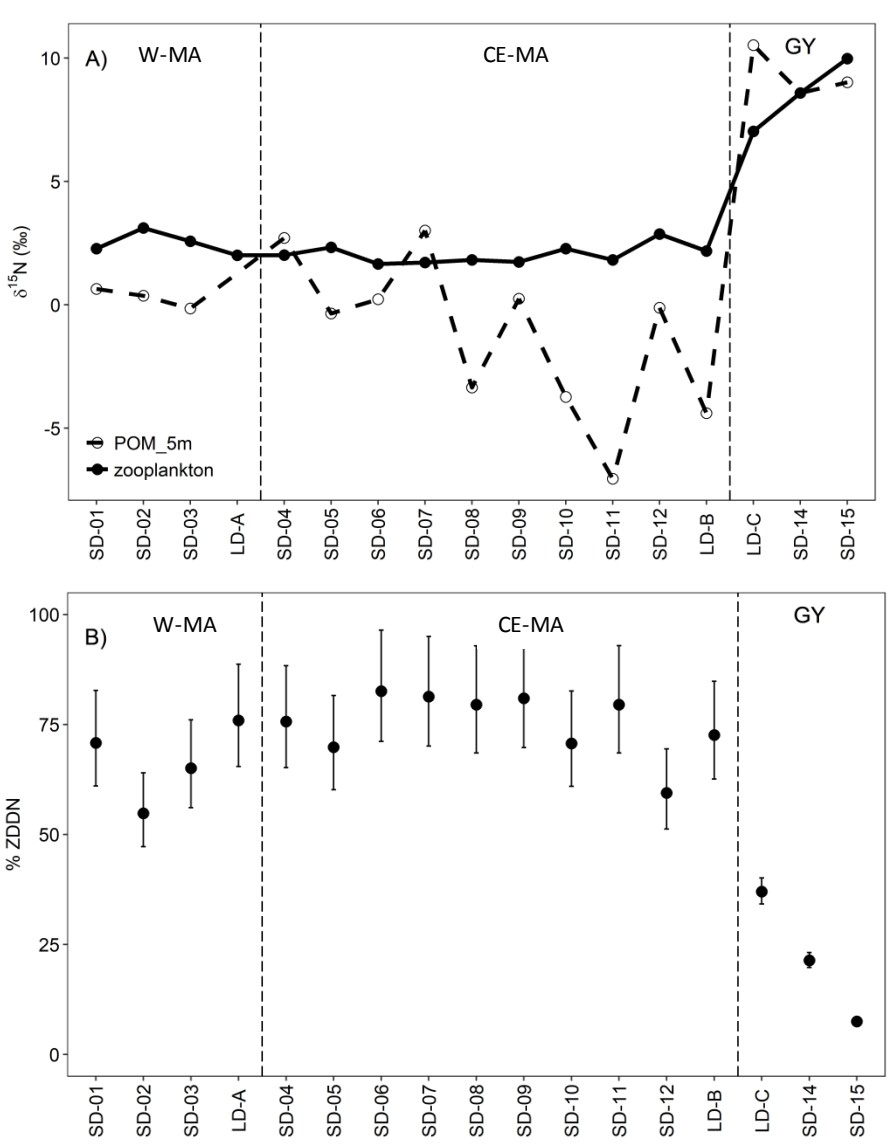

5 **Figure 11** A, B