# Peer review of "Mesozooplankton structure and functioning in the Western Tropical South Pacific along the 20° parallel south during the OUTPACE survey (February –April 2015)"

_Biogeosciences, 2017_

## Referee Comment (RC1) · Anonymous Referee #1 · 7 May 2018

General comments

This manuscript describes a number of measurements characterizing mesozooplankton communities along a productivity gradient in the tropical Pacific. It contains a significant number of new observations that will help to understand trophic dynamics and biogeochemical fluxes in this region. At first the manuscript seems inappropriate for Biogeochemistry, as the objectives are formulated as pure descriptions of mainly the description of the taxonomical composition and biomass of the communities, with only minor part dealing with biogeochemical fluxes. However, this manuscript apparently

contains results supporting other accompanying manuscripts derived from the same cruise and more focused in biogeochemistry. Only for this reason the manuscript could be accepted for publication in the same journal but after fixing several issues detailed below.

Specific comments

1) The present style is purely descriptive. The objectives are formulated as mere descriptions of zooplankton communities along a transect. There are no explicit hypotheses behind their formulation. The need for more data in the study region is a poor justification for attracting readers in this journal. This is reflected in a long abstract ending without a clear conclusion. In addition, these valuable data need to be accessible to other future users by storage in a data repository (e.g. PANGAEA). The authors need to consider this later point and add the appropriate reference to data storage in the revised version. 2) Because of the descriptive conception of the manuscript the writing is wordy, with a poor synthesis reflected by a large number of tables and figures in the main text. There is an unbalanced treatment of the objectives: much detail in the description of zooplankton communities (5 tables and 9 figures) but only one table and one figure to present the results for the second objective. This treatment confounds the reader and loses the focus on the implication of the different composition of the communities for the biogeochemistry of this region. The authors must consider reducing the description of the communities to a lower number of tables and figures. For instance focusing in multivariate analyses and leaving complementary indicators (as rank and diversity index) to supplementary materials, will help to understand the second objective. In addition, there are some results not clearly justified from the beginning. For instance, the record of zooplankton swimmers in the traps seems a bit odd in a general description of communities (unless it is used as an indicator of the migratory activity or of the potential for degradation of the sedimented matter). 3) The overall style of the manuscript indicates careless writing, with a number of mistakes and poor editing. This poor presentation greatly difficult the review process and affects

the understanding of the authors' interpretations. Particular attention must be taken with the use of acronyms (requiring definition at first use) and species names (see some specific corrections required below). 4) Methods (p 5, L 18): indicate sampling time (day or night) for the "regular" zooplankton stations. This is an important information as changes in day and night abundance and biomass have been found in the 5d-sampled stations. 5) Methods (p 7, L 7). Indicate analytical and measurement error for isotopic determinations. 6) Methods (p7, L 8). Indicate type of filters and volume filtered for seston determinations. 7) Methods (p7, L 28): define UVP (CTD may be acceptable without definition because of generalized use) 8) Methods (p8, L 12). Indicate the methods used for determining C, N, and P in zooplankton samples. 9) Methods (P9, L 8): why using only these variables for the PCA. Where other variables (e.g. nutrient concentrations) available? 10) Methods (p9, L 15) and thereafter: Spearman rank correlations are generally expressed with the Greek letter rho (ïĄš). I suggest using this letter instead of "Rs". 11) Results (P10, L10). I assume that differences between means were first studied by ANOVA as described but later paired differences were analysed with some kind of 'a posteriori' test. Indicate the type of test used and mark significant means in Table 1 for clarity (e.g. with different letters). 12) Results (P11, L6 and thereafter). Mean values and variability are mentioned several times in the text. In some cases the variability is defined as SD (standard deviation). I suggest defining this form in the first use and then use always the same format (mean±sd). Take into account that SD is also part of the code of some stations and its continued use in the text may confound the reader (e.g. P11, L31). 13) Results (P12, L29): use only full genus and species names at the first apparition in the text. Macrosetella gracilis is first cited in P 12, L17. Therefore it must be cited as M. gracilis thereafter (e.g. P12, L 29). Check that all species are cited in this way through the text. 14) Results (P13, L9) and Methods (P9, L24). Why using multiple regression to link environmental variables to NMDS first two dimensions? Justify the use of this method in preference to other alternatives (e.g. the BEST procedure in PRIMER V6). 15) Results (P14, L5-20). Consider expanding the description of the results related to the trophic interaction between phyto- and zooplankton, as this section appears to be the main link between the related manuscripts of the same cruise and contains the main biogeochemically relevant results. At the same time avoid repeating the text of the table heading in the main text (P14, L11-13) and use subscripts and superscripts for ammonium and phosphate (P14, L11). 16) Discussion. Consider reducing the length of the section dedicated to the description of communities (section 4.2) and, in general the titles of the subsections (e.g. 4.1. Characterization of biogeochemical regions; 4.2. Bottom-up control of zooplankton communities, 4.4. Top-down control of zooplankton on phytoplankton. 17) Discussion (P15, L22-23): Rephrase the description of correlations. Use "positive" and "negative" (instead of "good" and "inverse"). 18) Discussion (P15, L24-34). Explain better the causes of the change in the correlations between Chla and zooplankton variables. Only the eddy dynamics affect to the mismatch between phyto- and zooplankton? Consider also the different turnover time of phyto and zooplankton organisms (i.e. zooplankton integrate over longer periods). 19) Discussion (P16, L 12). Here is the first time that the study of swimmers is justified as an indicator of activity. It would be appropriate to state this justification earlier in the manuscript (e.g. in the introduction). 20) Discussion (P19, L10-13). Confuse and repetitive sentence. Rephrase to clarify the meaning: covariation of Chla with both N2-fixation and zooplankton variables suggest a link of N2-fixation with zooplankton. Also in P 19, L19: ". . .correlations between key species and diazotroph distributions. . ." 21) Discussion (P19, L21-34). All the trophic interpretation of the link between zooplankton consumers and N-fixers is made by direct grazing of filaments or particles. However, zooplankton can acquire diazotrophic N through microbial food webs, as the excreted DON can be taken up by bacteria, subsequently consumed by protozoans and metazoans ( Mulholland, 2007), as interpreted in other studies (e.g. McCarthy et al. 2007; Mompeán et al., 2013). 22) Discussion (P20, L1): remove italics for Thecosomata. 23) Discussion (P20, L17) define DDA 24) Discussion (section 4.4). I find this the most interesting part of the manuscript, dealing with the top-down effect of zooplankton on the primary production. However several key issues were not mentioned. For instance, does the estimates of zooplankton grazing match with the export (measured by the traps)? Is feasible to measure any export when zooplankton consumption accounts by >100% of primary production? Other important process to take into account is the zooplankton respiration. It can be assumed that zooplankton respiration would be high also when grazing and excretion is high, thus affecting the net carbon budget (P21, L1). Even when the estimations made from biomass and using equations from the literature (as in this case) only provide gross estimates of the real processes, they can be useful to detect future research needs and bottlenecks. A list of recommendations derived from the analysis of the fluxes in Table 7 would be appropriate. 25) The last sentence of the discussion (P21, L33-34) is not a conclusion and needs further clarification (tuna marine food web?). Because of the large number of results presented and discussed the final section of the manuscript would benefit from having the main conclusions summarized in a synthetic way. For instance, bottom-up and top-down control variability in the different regions.

Additional references: McCarthy, M.D., Benner, R., Lee, C., Fogel, M.L., 2007. Amino acid nitrogen isotopic fractionation patterns as indicators of heterotrophy in plankton, particulate, and dissolved organic matter. Geochimica et Cosmochimica Acta 71, 4727-4744. Mompeán, C., Bode, A., Benítez-Barrios, V.M., Domínguez-Yanes, J.F., Escánez, J., Fraile-Nuez, E., 2013. Spatial patterns of plankton biomass and stable isotopes reflect the influence of the nitrogen-fixer Trichodesmium along the subtropical North Atlantic. J. Plankton Res. 35(3), 513-525. Mulholland, M.R., 2007. The fate of nitrogen fixed by diazotrophs in the ocean. Biogeosciences 4(1), 37-51.

---

## Referee Comment (RC2) · Anonymous Referee #2 · 25 Jun 2018

GENERAL COMMENTS The present manuscript is part of the OUTPACE Experiment, a multidisciplinary effort to study the functioning of the western tropical South Pacific ecosystems and associated biogeochemical cycles. In that sense, the work presented by Carlotti et al. matches the scope of Biogeochemistry, since it includes the description of the mesozooplankton compartment as part of the studied ecosystems. It presents valuable information about mesozooplankton abundance, diversity and biomass, including a stable isotope analysis and estimations of carbon demand, grazing impact and zooplankton excretion rates in a poorly studied area, adding value

to the results presented here. However, the manuscript is often too descriptive, relying excessively on other analyses included within the same Special Issue and in other previous studies, masking the meaning of the present dataset. I think that the manuscript could be accepted for publication in Biogeosciences but only after major revisions.

SPECIFIC COMMENTS Grammar mistakes and poor editing are evident throughout the whole manuscript, while reading is difficult because of wordiness. Hypothesis are missing, and conclusions are not clear for the reader. I strongly recommend 1) reducing and rewriting the Discussion section, focusing on the results from this study, and also 2) balancing the story as well as the number of tables (e.g. including some of the latter as supplementary material). Some detailed comments: Abstract: (P1, L25) It would be more accurate to use "secondary consumers", rather than "mesozooplankton". (P1, L25-29) Please split up this sentence in two. (P2, L20). Please correct ingestion rates units. (P2, L21 and throughout the text) NH4+ and PO43- are a charged cation and anion, respectively; please correct. It is difficult to extract the main conclusions of the study from the Abstract. Introduction: (P3L5) This is the first time that the authors name ENSO, please define the acronym as El Niño Southern Oscillation (ENSO) here and not in Section 2.1. (P3L13) Please provide more details about the filamentous cyanobacteria biomass after summer blooms or link this paragraph with the following one. (P3L16) when referring to "productivity of zooplankton", do the authors refer to an increase in zooplankton biomass? Please correct. (P2L21-22). I assume there are some brackets missing here. (P4L10) Authors do not use quotation marks but the Spanish "ñ" when referring to El Niño, please be consistent when referring to La Niña. Material & Methods: (P5L2-9) Authors refer to Table 1 from Moutin et al (2017) for all general characteristics of the stations. However, a list of acronyms and main environmental features that could be relevant for the present zooplankton study would help the reader in a substantial way. (P5L17) Authors mention that station SD-13 was not sampled for zooplankton. Any reason for that? Please specify. (P5L22 and throughout the text) Please correct units and be consistent. In this case, the correct for would be m s-1. (P5L25) I guess that something is missing here, do you mean 0.3 m3 rev-1?
Please correct. (P5L30) Please rewrite; do not use symbols (+) in the description and include a formula for the sake of clarity. (P6L6 and throughout the text) Please correct to ind m-3. (P6L6) Why to use the Shannon-Weaver diversity index amongst others to estimate zooplankton diversity? Please provide a short explanation. (P6L23) Please add the word "software" after "Identifier". (P7L17-24) Please split up this paragraph in two sentences. (P8L8) Results: (P11L1). Chaetognaths are considered as gelatinous zooplankton, so it is wrong to consider this group apart during the analyses. Same is valid for Fig3B and Fig 9A. Unless there is a reason to consider chaetognaths separately —in that case, please specify— please correct this point throughout your manuscript. (P11L5) I think that authors refer to early life stages, rather than larval forms. Copepod larval forms are nauplii, while copepodites are copepod juveniles, both with their corresponding stages depending on the copepod species. Please differentiate both properly and correct accordingly in this paragraph and throughout the text. Discussion: (P17L14) Which group do the authors refer to when using the term "small zooplankton"? Please clarify. (P20L1-4) Why do authors refer here to the study from Caffin et al. (2017) and not to their own dataset (Fig 10)? Please correct accordingly. (P20L29-34) The fact that daily grazing pressure of zooplankton represents >100% (234%) of primary production calls for an argumentation of this result. According to the authors, which are the reasons of such a difference between their result with those from Dam et el. (1995)? (P21L33-34) This is the first time that authors mention the (possible) trophic link between the plankton community studied along the

manuscript and the tuna marine food web and needs clarification.

---

## Author Comment (AC1) · 4 Sep 2018

**General comments**

This manuscript describes a number of measurements characterizing mesozooplankton communities along a productivity gradient in the tropical Pacific. It contains a significantnumber of new observations that will help to understand trophic dynamics andbiogeochemical fluxes in this region. At first the manuscript seems inappropriate forBiogeochemistry, as the objectives are formulated as pure descriptions of mainly thedescription of the taxonomical composition and biomass of the communities, with onlyminor part dealing with biogeochemical fluxes. However, this manuscript apparentlycontains results supporting other accompanying manuscripts derived from the samecruise and more focused in biogeochemistry. Only for this reason the manuscript could be accepted for publication in the same journal but after fixing several issues detailedbelow.

**Specific comments**

1) The present style is purely descriptive. The objectives are formulated as mere descriptionsof zooplankton communities along a transect. There are no explicit hypothesesbehind their formulation. The need for more data in the study region is a poorjustification for attracting readers in this journal. This is reflected in a long abstractending without a clear conclusion.

*Answer*

*Abstract has been completely rewritten, presenting the main results of the different objectives which have been better balanced between trophic and biogeochemical processes and zooplankton community structure.*

In addition, these valuable data need to be accessibleto other future users by storage in a data repository (e.g. PANGAEA). The authorsneed to consider this later point and add the appropriate reference **to data storage** inthe revised version.

*Answer*

*Data will be stored in the data base of the INSU-CNRS cruise data base.* [http://www.obs-vlfr.fr/proof/php/outpace/outpace.php](http://www.obs-vlfr.fr/proof/php/outpace/outpace.php).

2) Because of the descriptive conception of the manuscript thewriting is wordy, with a poor synthesis reflected by a large number of tables and figuresin the main text. There is an unbalanced treatment of the objectives: much detail inthe description of zooplankton communities (5 tables and 9 figures) but only one tableand one figure to present the results for the second objective. This treatment confounds the reader and loses the focus on the implication of the different compositionof the communities for the biogeochemistry of this region. The authors must consider reducing the description of the communities to a lower number of tables and figures.For instance focusing in multivariate analyses and leaving complementary indicators (as rank and diversity index) to supplementary materials, will help to understand the second objective.In addition, there are some results not clearly justified from the beginning.For instance, the record of zooplankton swimmers in the traps seems a bit odd in a general description of communities (unless it is used as an indicator of the migratory activity or of the potential for degradation of the sedimented matter).

*Answer*

*As already mentioned, the different objectives have been rebalanced between trophic and biogeochemical processes and zooplankton community structure. Therefore, we strongly reduced the description of zooplankton community structure with reduction and recomposition of figures and tables, and focusing more on the multivariate analyses.The two other objectives on the Interactions with diazotroph microplankton and on the fluxes related to zooplankton have been developed with new data analysis synthesized in tables.*

*We agree on the referee's comment on the swimmers in sediment trap which is often very difficult to interpret. However, we kept the data from Caffin et al. (2018) to compare to our estimated mesozooplankton rates.*

3) Theoverall style of the manuscript indicates careless writing, with a number of mistakesand poor editing. This poor presentation greatly difficult the review process and affects the understanding of the authors' interpretations. Particular attention must be takenwith the use of acronyms (requiring definition at first use) and species names (seesome specific corrections required below).

**Answer**

*We hope that the new structuration of the manuscript make the paper easier for reading. We have been careful to have acronyms and species names correctly defined and spelled. The text has been read by a native speaker. However, we are aware that some sentences should still be reworded*

**Methods**

4) Methods (p 5, L 18): indicate samplingtime (day or night) for the "regular" zooplankton stations. This is an important informationas changes in day and night abundance and biomass have been found in the5d-sampled stations.

**Answer**

*It has been added in the text.*

5) Methods (p 7, L 7). Indicate analytical and measurement error for isotopic determinations.

**Answer**

*This point has been added in the text.*

*Stable nitrogen isotope analysis was performed with an Integra CN, SerCon Ltd. EA-IRMS. δ15N values were determined in parts per thousand (‰) relative to the external standard of atmospheric N. Repeated measurements of an internal standard indicated measurement precision of ± 0.13 ‰ for $\delta^{15}N$."*

6) Methods (p7, L 8). Indicate type of filters and volume filtered for seston determinations.

**Answer**

*For POM analyses, water samples were collected in 4.4L polycarbonate bottlesat depth corresponding to 50% and 1% of light attenuation. The samples were immediately filtered on pre-burnt (450 ° C, 4 h) 25 mm GFF filters and then analyzed by mass spectrometry for the determination of delta 15N naturalness*

*It has been added in the text.*

7) Methods (p7, L 28): define UVP (CTD may be acceptablewithout definition because of generalized use)

**Answer**

*It has been added in the text.*

8) Methods (p8, L 12). Indicate the methods used for determining C, N, and P in zooplankton samples.

**Answer**

*We quote the reference (Caffin et al., 2018a) where the methods are presented.*

*They wrote "Swimmers were both weighted and analyzed separately on EA-IRMS (Integra2, Sercon Ltd) to quantify exported PC and PN. Particulate phosphorus (PP) was analyzed by colorimetric method (880 nm) after mineralization according to Pujo-Pay and 15 Raimbault (1994)."*

9) Methods (P9, L 8): why using only these variables for the PCA. Where other variables (e.g.nutrient concentrations) available?

**Answer**

*This point has been added in the text.*
*We only considered variables pertinent for defining zooplankton habitat (temperature, salinity, food concentration)*

10) Methods (p9, L 15) and thereafter: Spearmanrank correlations are generally expressed with the Greek letter rho (ïA¸š). I suggest using this letter instead of "Rs".
***Answer***
*It has been changed in the text.*

**Results**
11) Results (P10, L10). I assume that differencesbetween means were first studied by ANOVA as described but later paired differences were analysed with some kind of 'a posteriori' test. Indicate the type of test used andmark significant means in Table 1 for clarity (e.g. with different letters).
***Answer***
*Post hoc Sheffé test were used to compare the paired difference between zones or LD stations. We have added this information in the method section. An also we have added letters in Tables 1, 2 and 3 to indicate homogeneous groups between zones or LD stations for the different variables.*

12) Results(P11, L6 and thereafter). Mean values and variability are mentioned several times inthe text. In some cases the variability is defined as SD (standard deviation). I suggestdefining this form in the first use and then use always the same format (mean_sd).Take into account that SD is also part of the code of some stations and its continueduse in the text may confound the reader (e.g. P11, L31).
***Answer***
*In the text, we now present mean values ± standard deviation. This presentation is defined in the legend of the tables.*

13) Results (P12, L29): useonly full genus and species names at the first apparition in the text. Macrosetella gracilisis first cited in P 12, L17. Therefore it must be cited as M. gracilis thereafter (e.g.P12, L 29). Check that all species are cited in this way through the text.
***Answer***
*We followed your recommendation in the new version.*

14) Results(P13, L9) and Methods (P9, L24). Why using multiple regression to link environmentalvariables to NMDS first two dimensions? Justify the use of this method in preference toother alternatives (e.g. the BEST procedure in PRIMER V6).
***Answer***
*Thanks for this remark; we applied the BEST procedure (we did not know before) which is more adapted than the multiple regression. We have modified the text accordingly*

15) Results (P14, L5-20).Consider expanding the description of the results related to the trophic interaction between phyto- and zooplankton, as this section appears to be the main link between therelated manuscripts of the same cruise and contains the main biogeochemicallyrelevantresults
***Answer***
New analyses have been done and the part has been rewritten. We added information on respiration, vertical flux and grazing pressure on phytoplankton size classes

At the same time avoid repeating the text of the table heading in the maintext (P14, L11-13) and use subscripts and superscripts for ammonium and phosphate(P14, L11).
***Answer***
*It has been changed in the text.*

**Discussion**

16) Discussion. Consider reducing the length of the section dedicated tothe descriptionof communities (section 4.2) and, in general the titles of the subsections(e.g. 4.1. Characterization of biogeochemical regions; 4.2. Bottom-up controlof zooplankton communities, 4.4.Top-down control of zooplankton on phytoplankton.

*Answer*

*We reduced the section 4.2 and we wrote shorter subtitles*

17) Discussion (P15, L22-23): Rephrase the description of correlations. Use "positive"and "negative" (instead of "good" and "inverse").

*Answer*

*We changed it.*

18) Discussion (P15, L24-34). Explainbetter the causes of the change in the correlations between Chla and zooplankton variables.Only the eddy dynamics affect to the mismatch between phyto- and zooplankton?Consider also the different turnover time of phyto and zooplankton organisms (i.e.zooplankton integrate over longer periods).

*Answer*

*We added a sentence to argue for quick zooplankton turnover time linked to high tropical temperature*

19) Discussion (P16, L 12). Here is the first time that the study of swimmers is justified as an indicator of activity. It would be appropriate to state this justification earlier in the manuscript (e.g. in the introduction).

*Answer*

*Now "swimmers" are presented in the M&M part 2.7.*

20)Discussion (P19, L10-13). Confuse and repetitive sentence. Rephrase to clarify the meaning: covariation of Chla with both N2-fixation and zooplankton variables suggesta link of N2-fixation with zooplankton. Also in P 19, L19: ": : :correlations between keyspecies and diazotroph distributions: : :"

*Answer*

*The discussion has been fully rewritten*

21) Discussion (P19, L21-34). All the trophicinterpretation of the link between zooplankton consumers and N-fixers is made by directgrazing of filaments or particles. However, zooplankton can acquire diazotrophic N through microbial food webs, as the excreted DON can be taken up by bacteria, subsequentlyconsumed by protozoans and metazoans ( Mulholland, 2007), as interpretedin other studies (e.g. McCarthy et al. 2007; Mompeán et al., 2013).

*Answer*

*The indirect link between diazotroph microorganisms and zooplankton is now discussed in the new version.*

22) Discussion (P20, L1): remove italics for *Thecosomata*.

*Answer*

*We changed it.*

23) Discussion (P20, L17) define DDA

*Answer*

*We defined it in the introduction*

24) Discussion (section 4.4). I find this the most interesting part of the manuscript, dealingwith the top-down effect of zooplankton on the primary production. However severalkey issues were not

mentioned. For instance, does the estimates of zooplankton grazing match with the export (measured by the traps)?

Is feasible to measure any exportwhen zooplankton consumption accounts by >100% of primary production?

*Answer*

*In the revised version we compare our ingestion estimates to the particle carbon export (from Caffin et al 2018) highlighting that only a small fraction of unassimilated ingestion (fecal pellets) would contribute to the material collected in the traps, which goes in the sense of a unbalance between new production and export highlighted by these authors*

24) Other importantprocess to take into account is the zooplankton respiration. It can be assumedthat zooplankton respiration would be high also when grazing and excretion is high,thus affecting the net carbon budget (P21, L1). Even when the estimations made frombiomass and using equations from the literature (as in this case) only provide grossestimates of the real processes, they can be useful to detect future research needsand bottlenecks.

*Answer*

*Respiration rates have been calculated and discussed*

A list of recommendations derived from the analysis of the fluxes inTable 7 would be appropriate.

*Answer*

*This table has been deeply amended (with new analyses of the data) to better focus on trophic and biogeochemical aspects in the revised version, as recommended by both referees. This complementary information is now more thoroughly discussed*

25) The last sentence of the discussion (P21, L33-34) isnot a conclusion and needs further clarification (tuna marine food web?).

*Answer:*

*This sentence has been shifted in the introduction of the paper, as it is a key characteristic of the region. Our study is based on mesozooplankton and we did not study any linkages with the mesopelagic fish.*

*One main output of the paper is to highlight quite high rates of phyto and zooplankton production (despite rather low stocks) which may explain a consistent trophic flux up to tunas.*

Because of the large number of results presented and discussed the final section of the manuscriptwould benefit from having the main conclusions summarized in a synthetic way. For instance, bottom-up and top-down control variability in the different regions.

*Answer*

*We have added a small conclusive paragraph highlight and try to interpret the high variability and high values of the top down (ingestion) and bottom-up (excretion) impacts and of the carbon fluxes associated to zooplankton observed in our study*

**Additional references:**

McCarthy, M.D., Benner, R., Lee, C., Fogel, M.L., 2007. Aminoacid nitrogen isotopic fractionation patterns as indicators of heterotrophy in plankton, particulate, and dissolved organic matter. Geochimica et Cosmochimica Acta 71, 4727-4744.

Mompeán, C., Bode, A., Benítez-Barrios, V.M., Domínguez-Yanes, J.F.,Escánez, J., Fraile-Nuez, E., 2013. Spatial patterns of plankton biomass and stableisotopes reflect the influence of the nitrogen-fixer *Trichodesmium* along the subtropical North Atlantic. J. Plankton Res. 35(3), 513-525.

Mulholland, M.R., 2007. The fate of nitrogen fixed by diazotrophs in the ocean. Biogeosciences 4(1), 37-51.

---

## Author Comment (AC2) · 4 Sep 2018

**GENERAL COMMENTS**

The present manuscript is part of the OUTPACE Experiment,a multidisciplinary effort to study the functioning of the western tropical SouthPacific ecosystems and associated biogeochemical cycles. In that sense, the work presented by Carlotti et al. matches the scope of Biogeochemistry, since it includesthe description of the mesozooplankton compartment as part of the studied ecosystems.It presents valuable information about mesozooplankton abundance, diversity and biomass, including a stable isotope analysis and estimations of carbon demand,grazing impact and zooplankton excretion rates in a poorly studied area, adding valueto the results presented here. However, the manuscript is often too descriptive, relying excessively on other analyses included within the same Special Issue and in other previous studies, masking the meaning of the present dataset. I think that the manuscript could be accepted for publication in Biogeosciences but only after major revisions.

**SPECIFIC COMMENTS**
Grammar mistakes and poor editing are evident throughout the whole manuscript, while reading is difficult because of wordiness.

*Answer:*
*We hope that the new structuration of the manuscript make the paper easier for reading. The text has been read by a native speaker.*

Hypothesis are missing, and conclusions are not clear for the reader.
I strongly recommend
   1) reducing and rewriting the Discussion section, focusing on the results from this study, and also

2) balancing the story as well as the number of tables (e.g. including some of the latteras supplementary material).
*Answer*:
*The different objectives have been rebalanced between trophic and biogeochemical processes and zooplankton community structure. Therefore, we strongly reduced the description of zooplankton community structure with reduction and re composition of figures and tables. The two other objectives on the Interactions with diazotroph microplankton and on the fluxes related to zooplankton have been developed with new data analysis synthesized in tables.*
*The discussion part is structured in relation with these three objectives.*

**Some detailed comments:**

**Abstract:**
(P1, L25) It would be more accurate to use "secondary consumers", rather than "mesozooplankton".
*Answer:*
*We considered now herbivorours mesozooplankton and carnivorous zooplankton  from the observed taxa. Biomasses of both groups have been separated to estimate the impact on phytoplankton.*

(P1,L25-29) Please split up this sentence in two.
***Answer***:
*We changed it.*

(P2, L20). Please correct ingestion rates units.
***Answer***:
*We changed it.*

(P2, L21 and throughout the text) NH4+ and PO43☐are a charged cation and anion, respectively; please correct.
***Answer***:
*We changed it.*

It is difficult to extract the main conclusions of the study from the Abstract.
***Answer***:
*The abstract has been fully rewritten.*

**Introduction:**
(P3L5) This is the first time that the authorsname ENSO, please define the acronym as El Niño Southern Oscillation (ENSO) here and not in Section 2.1.
***Answer***:
*We put the definition of ENSO in the introduction part.*

(P3L13) Please provide more details about the filamentous cyanobacteria biomass after summer blooms or link this paragraph with the following one.
***Answer***:
*We linked the two paragraphs.*

(P3L16) when referring to "productivity of zooplankton", do the authors refer toan increase in zooplankton biomass? Please correct.
***Answer***:
*You are right. We changed it.*

(P2 L21-22). I assume thereare some brackets missing here.
***Answer***:
*It has been corrected.*

(P4L10) Authors do not use quotation marks but theSpanish "ñ" when referring to El Niño, please be consistent when referring to La Niña.
***Answer***:
*It has been corrected.*

**Material & Methods:**
(P5L2-9) Authors refer to Table 1 from Moutin et al (2017) for all general characteristics of the stations. However, a list of acronyms and main environmental features that could be relevant for the present zooplankton study would help the reader in a substantial way.
***Answer***:
Moutin et al. (2017) ' Table 1 present the date, location, and general characteristics of the stations investigated along the OUTPACE transect : Distance in kilometers from the first SD station (SD1),

Arrival date, Departure date, Latitude, Longitude, Bottom depth. We believe that the main environmental features needed for the best interpretation of our results have been presented. We pay attention that the different acronyms have been correctly explained when firstly quoted.

(P5L17) Authors mention that station SD-13 was not sampled for zooplankton. Any reason for that? Please specify.

***Answer:***
*Station SD13 was an additional very short station done just out of the bloom patch, not initially planned and with limited measurements. Zooplankton net tows were not realized at this station.*

(P5L22 and throughout the text) Please correct units and be consistent. In this case, the correct for would bem s-1. (P5L25) I guess that something is missing here, do you mean 0.3 m3 rev-1? Please correct.

***Answer:***
*We give more details in the text. R: is revolution and there are 10 counts per revolution. K units are correct.*

(P5L30) Please rewrite; do not use symbols (+) in the description andinclude a formula for the sake of clarity.

***Answer:***
*It has been corrected.*

(P6L6 and throughout the text) Please correct to ind m-3.

***Answer:***
It has been corrected.

(P6L6) Why to use the Shannon-Weaver diversity index amongst others to estimate zooplankton diversity? Please provide a short explanation. (P6L23)

***Answer:***
Shannon-Weaver diversity index is a widely used method of calculating biotic diversity in plankton studies.

Please add the word "software" after "Identifier".

***Answer:***
*It has been added*

(P7L17-24) Please split up this paragraph in two sentences. (P8L8)

***Answer:***
*It has been rephrased in three sentences !*

**Results:**
(P11L1). Chaetognaths are considered as gelatinouszooplankton, so it is wrong to consider this group apart during the analyses. Same is valid for Fig3B and Fig 9A. Unless there is a reason to consider chaetognaths separately ˘A ˘ T in that case, please specify ˘A˘T please correct this point throughout yourmanuscript.

**Answer:**
In the table, chaetognaths are now grouped to gelatinous.  Fig3B and Fig 9A have been recomposed.

(P11L5) I think that authors refer to early life stages, rather than larvalforms. Copepod larval forms are nauplii, while copepodites are copepod juveniles,both with their corresponding stages depending on the copepod species. Please differentiateboth properly and correct accordingly in this paragraph and throughout thetext.

*Answer:*
*It has been changed*

**Discussion:**
(P17L14) Which group do the authors refer to when using the term"small zooplankton"? Please clarify.
*Answer:*
*In the new version this part has been removed*

(P20L1-4) Why do authors refer here to the study from Caffin et al. (2017) and not to their own dataset (Fig 10)? Please correct accordingly.
**Answer:**
*It has been changed and we now refer to Fig.7 A and B.*

(P20L29-34) The fact that daily grazing pressure of zooplankton represents>100% (234%) of primary production calls for an argumentation of this result. According to the authors, which are the reasons of such a difference between their result with those from Dam et al. (1995)?
*Answer:*
*We now quote the Calbet (2001) paper who presents a synthetic analysis of grazing vs. primary production based on data compilation (including those by Dam et al. (1995).*

(P21L33-34) This is the first time that authors mention the (possible) trophic link between the plankton community studied along the manuscript and the tuna marine food web and needs clarification.
**Answer:**
*This sentence has been shifted in the introduction of the paper, as it is a key characteristic of the region. Our study is based on mesozooplankton and we did not study any linkages with the mesopelagic fish.*

*One main output of the paper is to highlight quite high rates of phyto and zooplankton production (despite rather low stocks) which may explain a consistent trophic flux up to tunas.*

---

## Author Comment (AC3) · 4 Sep 2018

Please find the new version of the ms in the attached file

Please also note the supplement to this comment:
https://www.biogeosciences-discuss.net/bg-2017-573/bg-2017-573-AC3-supplement.pdf

---

## Author Comment (AC4) · 4 Sep 2018

[revised manuscript text omitted]

**A**

[Figure]

**B**

**Figure 4A, B**

[Figure]

**Figure 5 A, B,C**

[Figure]

5   **Figure 6 A, B, C, D, E, F**

[Figure]

**Figure 7 A, B**

[Figure]

5    **Figure 8** A, B

**Supplementary table**

**Table S1** List of zooplanktonic taxa collected and identified during the 2015 OUTPACE cruise with average percentage of abundance within the 0-200 upper meters of the water column in the four Clusters defined in the PCA analysis on environmental variables. W-MA = Western Melanesian archipelago, CE-MA= Central and Eastern Melanesian archipelago, BL = station B, blooming conditions, and GY = subtropical gyre.

| | W-MA | CE-MA | BL | GY | | W-MA | CE-MA | BL | GY |
|---|---|---|---|---|---|---|---|---|---|
| **COPEPODS** | | | | | **COPEPODS (follow)** | | | | |
| Copepod nauplii | 10.02 | 8.96 | 9.63 | 9.22 | *Phaenna spinifera* | 0.01 | | | |
| Undetermined copepodites | 0.01 | | | | *Pleuromamma* | 0.09 | 0.72 | 0.81 | 1.64 |
| *Acartia* | 0.76 | 0.95 | 0.12 | 0.65 | *Sapphirina* | 0.05 | 0.06 | 0.05 | 0.10 |
| *Aetideus* | | 0.04 | | | *Scaphocalanus* | 0.01 | 0.03 | | |
| *Calanidae* | 1.44 | 1.70 | 1.73 | 1.25 | *Scolecithrix* | 0.09 | 0.26 | | 0.00 |
| *Calanopia* | 0.05 | 0.38 | 0.55 | 0.21 | *Scottocalanus* | | 0.00 | | |
| *Calanus* | 0.17 | 0.08 | 0.13 | 0.01 | *Subeucalanus / Eucalanu* | 0.09 | 0.01 | | |
| *Calocalanus* | 2.68 | 1.81 | 1.34 | 1.53 | *Temora* | 0.60 | 0.34 | 0.33 | 0.07 |
| *Candacia* | 0.27 | 0.33 | 0.36 | 0.42 | **OTHER CRUSTACEANS** | | | | |
| *Centropages* | 0.05 | 0.03 | | 0.04 | Amphipoda | 0.02 | 0.01 | 0.15 | 0.04 |
| *Clausocalanus/Paracalanus* | 18.83 | 17.28 | 16.04 | 24.70 | Euphausiacea | 0.18 | 0.37 | 0.39 | 0.22 |
| *Clytemnestra* | 0.01 | 0.03 | 0.03 | | Lucifer | 0.01 | 0.06 | | |
| *Copilia* | 0.17 | 0.12 | 0.17 | 0.14 | Ostracoda | 2.01 | 2.38 | 4.26 | 2.22 |
| *Corycaeus* | 5.29 | 4.20 | 3.04 | 8.90 | Pseudoevadne | 0.04 | | | 0.34 |
| *Cosmocalanus darwini* | | | 0.05 | | **GELATINOUS** | | | | |
| *Ctenocalanus* | | 0.12 | | | Appendicularia | 15.00 | 13.34 | 11.90 | 11.44 |
| *Cyclopoida* | | 0.01 | | 0.03 | Doliole | 0.18 | 0.09 | 0.39 | 0.21 |
| *Eucalanus* | 0.08 | 0.17 | 1.23 | 0.33 | Salpidae | 0.14 | 0.10 | 0.08 | 0.30 |
| *Euchaeta/Paraeuchaeta* | 0.06 | 0.10 | 0.03 | 0.07 | Siphonophora | 0.24 | 0.26 | 0.14 | 0.24 |
| *Euterpina acutifrons* | 0.02 | 0.06 | | | Hydrozoa | 0.06 | 0.06 | 0.19 | 0.04 |
| *Haloptilus* | 0.25 | 0.30 | 0.56 | 0.33 | **CHAETOGNATHA** | 2.13 | 2.56 | 1.98 | 0.60 |
| *Harpacticoida* | 0.01 | | | | **MOLLUSCS** | | | | |
| *Heterorhabdus* | 0.05 | 0.08 | 0.05 | 0.14 | Thecosomata | 5.29 | 3.81 | 2.24 | 1.95 |
| *Lubbockia* | 0.10 | 0.07 | 0.24 | 0.10 | **MEROPLANKTON** | | | | |
| *Lucicutia* | 1.11 | 1.27 | 1.54 | 1.41 | Decapod larvae | 0.01 | 0.12 | | |
| *Macrosetella gracilis* | 0.11 | 0.05 | 0.73 | 0.03 | Cephalopod larvae | 0.02 | | | |
| *Mecynocera clausi* | 1.20 | 1.64 | 1.17 | 2.03 | Cirripedia larvae | | 0.01 | | |
| *Mesocalanus/Neocalanus* | 0.08 | 0.05 | 0.18 | 0.07 | Echinoderm larvae | | 0.02 | 0.03 | |
| *Microsetella* | 3.08 | 4.13 | 4.38 | 2.83 | Lamellibranch larvae | 0.17 | 0.41 | 0.12 | 0.32 |
| *Miracia efferata* | 0.01 | 0.05 | 0.08 | | Gasteropod larvae | | | 0.08 | |
| *Mormonilla/Neomormonilla* | 0.08 | 0.23 | 0.13 | 0.12 | Polychaets larvae | 0.28 | 0.21 | 0.51 | 0.43 |
| *Nannocalanus minor* | 0.32 | 0.36 | 0.05 | 0.04 | Teleostei eggs | | 0.06 | | |
| *Oithona* | 12.90 | 15.89 | 13.16 | 13.54 | Teleostei larvae | | 0.03 | 0.03 | 0.00 |
| *Oncaea* | 13.88 | 14.10 | 19.53 | 11.65 | Branchiostoma | 0.01 | | | |
| *Paracalanus* | 0.13 | 0.03 | 0.05 | 0.03 | Larvae unknown | 0.05 | 0.05 | 0.05 | 0.07 |

---

## Referee Report (RR1)

**GENERAL COMMENTS**

In the revised version of this manuscript, the authors were taken into consideration the comments and the manuscript has improved in coherence and readability. The paper is more balanced now, and the structure and study objectives are clearer for the reader. However, some reviews are still needed. The manuscript is still missing consistency and need editing work. Just some examples: inconsistency in taxonomic groups naming (English vs Latin names, uppercases use, etc), punctuation/spacing errors, differences in naming (e.g. Chla-*a* vs chla), omission of anions and cations' charges, missing the use of bold font when naming some figures and tables within the text, etc.

I only recommend the publication of the present manuscript after extensive proofreading and spellchecks.

Please find some remarks below:

**SPECIFIC COMMENTS**

P1L25: Please delete the word "south" — it is redundant after 20° S.

P2 and throughout the text: $NH_4^+$ and $PO_4^{3-}$ are a charged cation and an anion, respectively. Although the authors said in their responses to my comments that they had changed it, they actually didn't. Please correct.

P4L10: I realise you corrected the Ñ throughout the document after my previous comments. Please correct to "La Niña" also here.

P8L29: Please delete "a" — "a constant values" is grammatically incorrect.

P9L29: To avoid overworking and improve readability, please rewrite to " We estimated the potential contribution of zooplankton excretion to nitrogen and phosphorous requirements for phytoplankton from primary production using Redfields's ratios."

P10L5-6: Please reword for the sake of clarity.

P14L11: Please remind the reader which are the 3 long duration stations.

P19L5: Over worded; please rewrite.

P20L9: Please correct "that" (change to "than") and "considered" ("consider").

Table 1: Salinity should not have units. Please correct.

Table 5 (and throughout the whole manuscript): Appendicularia is not a species, same as nauplii, Thecosomata, Chaetognatha and Ostracoda. Please correct accordingly.

Fig 3B: Please use italics for all copepod species.

Table S1: Please correct "Doliole" to Doliolida. Chaetognaths should be within the Gelatinous Zooplankton group.

---

## Author Response (AR2)

**Answers to scientific editor and referees**

**Editor**

Abstract file
*Answer:* Corrections suggested by the editor in the abstract file have been taken into account.

Manuscript file :
*Answer:* Corrections suggested by the editor in the ms. file have been taken into account.

**Referee 1**
Suggestions for revision or reasons for rejection (will be published if the paper is accepted for final publication)
I have found one minor change required: the definition of the DDN was not in the introduction, as the authors mention in their response, but it is defined only in the abstract.

*Answer: Definition of DDN is now included in the introduction.*

**Referee 2**

Suggestions for revision or reasons for rejection (will be published if the paper is accepted for final publication)
Some examples of remarks that should be considered by the authors throughout the manuscript: inconsistency in taxonomic groups naming (English vs Latin names, uppercases use, etc), punctuation/spacing errors, differences in naming (e.g. Chla-a vs chla), omission of anions and cations' charges, missing the use of bold font when naming some figures and tables within the text, etc.

GENERAL COMMENTS
In the revised version of this manuscript, the authors were taken into consideration the comments and the manuscript has improved in coherence and readability. The paper is more balanced now, and the structure and study objectives are clearer for the reader. However, some reviews are still needed. The manuscript is still missing consistency and need editing work.

Just some examples:

inconsistency in taxonomic groups naming (English vs Latin names, uppercases use, etc), *Answer: We homogenized as well as possible the taxonomic groups naming in relation to table S1*
punctuation/spacing errors, *Answer: This has been homogenized*
differences in naming (e.g. Chla-a vs chla), : *Answer: This has been homogenized*
omission of anions and cations' charges, *Answer: Charges have been added for all quoted anions and cations*
missing the use of bold font when naming some figures and tables within the text, etc. *Answer: This has been homogenized*

I only recommend the publication of the present manuscript after extensive proofreading and spellchecks. *Answer: The ms has been checked by an native English speaker M. M. Paul.*

Please find some remarks below:

SPECIFIC COMMENTS

P1L25: Please delete the word "south" — it is redundant after 20° S. *Answer: "South" has been deleted*

P2 and throughout the text: NH4+ and PO43- are a charged cation and an anion, respectively. Although the authors said in their responses to my comments that they had changed it, they actually didn't. Please correct. *Answer: It has been changed at all places where these symbols occur*

P4L10: I realise you corrected the Ñ throughout the document after my previous comments. Please correct to "La Niña" also here. *Answer: "La Niña'event" has been corrected with the right editing*

P8L29: Please delete "a" — "a constant values" is grammatically incorrect. *Answer: "a" has been deleted*

P9L29: To avoid overworking and improve readability, please rewrite to " We estimated the potential contribution of zooplankton excretion to nitrogen and phosphorous requirements for phytoplankton from primary production using Redfields's ratios." *Answer: The sentence has been changed as requested.*

P10L5-6: Please reword for the sake of clarity. *Answer: The sentence has been simplified :" A Bray Curtis matrix 'species – stations' of square root transformed abundance data was used to estimate station similarity"*

P14L11: Please remind the reader which are the 3 long duration stations. *Answer: The abbreviations of stations have been added : "3 long duration stations (LD-A, LD-B and LD-C)"*

P19L5: Over worded; please rewrite. *Answer: The sentence as been splitted in two shorter sentences.*

P20L9: Please correct "that" (change to "than") and "considered" ("consider"). *Answer: These words have been corrected*

Table 1: Salinity should not have units. Please correct. *Answer:  Salinity unit has been deleted*

Table 5 (and throughout the whole manuscript): Appendicularia is not a species, same as nauplii, Thecosomata, Chaetognatha and Ostracoda. Please correct accordingly. *Answer: "Taxa" has been written instead of "species"*

Fig 3B: Please use italics for all copepod species. *Answer:  It has been changed with all copepod species oin italics*

Table S1: Please correct "Doliole" to Doliolida. Chaetognaths should be within the Gelatinous Zooplankton group. *Answer: The two corrections have been done in Table S1.*